# CAMELS-DK: Hydrometeorological Time Series and Landscape Attributes for 3330 Danish Catchments with Streamflow Observations from 304 Gauged Stations

Jun Liu[1], Julian Koch[1], Simon Stisen[1], Lars Troldborg[1], Anker Lajer Højberg[1], Hans Thodsen[2], Mark F. T. Hansen[1], Raphael J. M. Schneider[1]

[1]Department of hydrology, Geological Survey of Denmark and Greenland, Copenhagen, Denmark
[2]Department of Bioscience, Aarhus University, Silkeborg, Denmark

*Correspondence to*: Jun Liu (juliu@geus.dk)

**Abstract**

Large samples of hydrometeorological time series and catchment attributes are critical for improving the understanding of complex hydrological processes, hydrological model development and performance benchmarking. CAMELS (Catchment Attributes and Meteorology for Large-sample Studies) datasets have been developed in several countries and regions around

the world, providing valuable data sources and testbeds for hydrological analysis and new frontiers in data-driven hydrological modelling. Regarding the lack of samples from low-land, groundwater-dominated, small-sized catchments, we develop an extensive repository of a CAMELS-style dataset for Denmark (CAMELS-DK). This CAMELS addition is the first containing both, gauged and ungauged catchments as well as detailed groundwater information. The dataset provides dynamic and static variables for 3330 catchments covering all of Denmark from various hydrogeological datasets, meteorological observations,

and a well-established national-scale hydrological model. For 304 of those catchments, streamflow observations are provided, whereas simulated streamflow is provided for all 3330 catchments. The dataset contains timeseries spanning 30 years (1989-2019) with a daily timestep, and the data will be updated once new observations and model simulations become available. The dense and full spatial coverage for all 3330 catchments, instead of only gauged catchments, together with the addition of various simulation data from a distributed, process-based model enhance the applicability of such CAMELS data, for example,

for the development of data-driven and hybrid physical informed modelling frameworks or other cases where consistent full spatial coverage is required. We also provide quantities related to human impact on the hydrological system in Denmark, such as groundwater abstraction and irrigation. The CAMELS-DK dataset is freely available at https://doi.org/10.22008/FK2/AZXSYP (Koch et al., 2024).

# 1 Introduction

Hydrometeorological time series and catchment attributes are crucial for understanding and modelling hydrological systems (Andersson et al., 2015; McMillan et al., 2018). Long-term records of water cycle variables, such as precipitation, evapotranspiration (ET), streamflow, and groundwater levels, give profound insights into dynamics and trends of water movement. These records form the basis for supporting water resource management and climate change adaptation. Sufficient hydrological data are required by scientists and engineers to manage water resources effectively and to make accurate predictions of hydrological extremes (Van Loon, 2015). Catchment attributes provide information on the physical characteristics of catchments, such as topography, soil type, land use, and geology, which are important for understanding how catchments respond to meteorological events and are relevant for land-use planning and catchment management. The similarity of catchment attributes is useful in comparative hydrology studies, which facilitate the transfer of knowledge from data-rich to ungauged catchments or regions where direct hydrological measurements are scarce or non-existent (Sawicz et al., 2011; Singh et al., 2014; Tegegn et al., 2022; Tshimanga et al., 2022). Being able to predict in ungauged catchments enhances our ability to manage water resources efficiently and sustainably.

The availability of open-access environmental data is increasing, however, data sources are distributed across different platforms and stored in various formats, which requires further efforts in data collection and pre-processing. Many state authorities and governmental research institutes comply with open data policies, significantly breaking down the barriers related to data sharing issues that have existed in the hydrology community for decades (Kibler et al., 2014). Numerous platforms and websites offer hydrological data from various sources, which greatly benefits large-scale studies. For example, the Global Runoff Data Centre (GRDC) provides streamflow measurements from over 10,000 hydrological stations (GRDC, 2020), the HydroSHEDS database offers a suite of hydrographic data (Lehner et al., 2008), the European Centre for Medium-Range Weather Forecasts provides multiple climate reanalysis products (Hersbach et al., 2020), and Hydroweb supplies satellite-derived water levels for global rivers and lakes (Da Silva et al., 2010). However, large-scale studies face a key challenge: They require a vast amount of data for development, and users often spend considerable time and effort navigating through different platforms for data collection and employing various programming packages for data pre-processing.

Many studies are, consequently, focused on creating large-sample hydrology (LSH) datasets providing streamflow data for a large number of catchments following the Findable, Accessible, Interoperable, and Reusable (FAIR) principle (Wilkinson et al., 2016). Newman et al. (2015b) created a comprehensive hydrometeorological dataset, which includes daily forcings and hydrologic response data for 671 small- to medium-sized basins throughout the contiguous United States (CONUS). This dataset was further enriched by Addor et al. (2017), who coined the term "Catchment Attributes and Meteorology for Large-sample Studies" (CAMELS) to describe their collection of catchment attributes for the same basins. Since then, CAMELS datasets have been developed in several countries, including Chile (Alvarez-Garreton et al., 2018), Brazil (Chagas et al., 2020), Great Britain (Coxon et al., 2020), Australia (Fowler et al., 2021), and Switzerland (Höge et al., 2023). Some other datasets provide hydrometeorological timeseries and catchment characteristics similar to CAMELS conventions, but with diverging

naming, such as LArge-SaMple DAta for Hydrology and Environmental Sciences for Central Europe (LamaH-CE, Klingler et al., 2021) and Iceland (LamaH-Ice, Helgason and Nijssen, 2023), historical hydro-meteorological time series and signatures for 24 catchments in Haiti (Simbi, Bathelemy et al., 2023), a large-scale benchmark dataset for data-driven streamflow forecasting (WaterBench-Iowa, Demir et al., 2022), China Catchment Attributes and Meteorology dataset (CCAM, Hao et al., 2021). These datasets are open access, well-formatted, and encompass a wide range of comprehensive variables related to hydrological processes (Klingler et al., 2021). They provide information for hydrological studies and water resources management (Frame et al., 2021; Jehn et al., 2020; Meyer Oliveira et al., 2023; Tang et al., 2023), but also serve as reference for the development of hydrological models, as well as the training and testing of data-driven algorithms (Kratzert et al., 2018, 2019, 2021; Liu et al., 2023; Mai et al., 2022; Nearing et al., 2024; Yin et al., 2022). Details about recent progress and their applications have been summarized by Addor et al. (2020). Kratzert et al. (2023) introduced the Caravan platform, which consolidates the national CAMELS datasets into a singular dataset derived from global sources, which further improve the accessibility of the dataset. However, Clerc-Schwarzenbach et al. (2024) concerned the Caravan dataset, which uses ERA5-Land reanalysis data, for exhibiting an unrealistically high potential evapotranspiration and a significant discrepancy in precipitation relative to the original CAMELS datasets. Such differences in meteorological forcings influence model results. Consequently, they advocated for the augmentation of the Caravan dataset with the forcing data present in the original CAMELS dataset, underscoring the ongoing relevance of the original CAMELS development.

While the already existing CAMELS datasets cover a wide range of hydroclimatic conditions and catchments characteristics it can still be considered incomplete in terms of the catchment diversity. Especially a lack of samples from low-lying, small and groundwater-dominated catchments. In the existing CAMELS datasets with a total number of 3308 catchments (CH: 331, GB:671, CL:516, BR: 897, AUS:222, US: 671), only 7.5% of the catchments (250 out of 3308) have an elevation lower than 100 m, and 9.5 % of the catchments have an area size smaller than 50 km$^2$. Hydrological regimes, with respect to their baseflow and peak flow differ between small and large catchments. The same holds for low lying catchments, where hydrological regimes are more affected by groundwater related processes. Hence, samples of small-sized and groundwater dominated catchments are necessary to increase the diversity of current existing CAMELS family. Additionally, information and data from well-established physical hydrological models have not yet been provided by previous LSH datasets. Simulated runoff, obtained from hydrological models, offers a valuable benchmark dataset, and can potentially also be provided at ungauged basins.

It is worthy to mention that LSH provides an invaluable foundation to scientists aiming to apply data-driven machine learning (ML) techniques in hydrology, as they undoubtedly offer an ideal environment for benchmarking, training, and testing ML algorithms. CAMELS and similar LSH have already contributed to ML advancements, such as predicting streamflow (Wilbrand et al., 2023), transfer learning (Ma et al., 2021), testing advanced algorithms (Yin et al., 2022), hybrid modelling (Espinoza et al., 2023) and global-scale flood forecasting (Nearing et al., 2024). In the context of ML-based hydrological modelling, physical-informed data driven algorithms, which combine the strength of traditional physics-based approaches with

ML models, show often enhanced performance(Konapala et al., 2020). Therefore, hydrological information from well-established hydrological models (if existing), would further benefit applications of CAMELS datasets. Previous CAMELS datasets provide simulated streamflow from conceptual hydrological models, such as LamaH-CE, which is insufficient in many Danish River systems. Liu et al. (2024) tested several hybrid schemes of combining simulations from a physically based hydrological model (PBM) and climate variables for streamflow estimation in Denmark, and the study found that the

combination of PBM simulations (such as shallow groundwater levels, soil water content, streamflow) and climate variables achieve better performance with a ML model.

The points presented above have motivated us to compile and introduce CAMELS-DK for Danish catchments. Denmark covers roughly 43,000 km$^2$, the topography is flat (highest point is 170 m above sea level), climate is temperate with precipitation evenly distributed over the year (annual average precipitation ranging between 600 mm in the east to 1000 mm in the west).

The flat terrain and wet climate generate about 69,000 km of river courses and 195,000 lakes (Danish Environmental Protection Agency, 2022). Groundwater levels close to ground surface are found throughout the country, reliable surface water modelling in Denmark always requires information of groundwater dynamics (Duque et al., 2023; Koch et al., 2021; Schneider et al., 2022). Ignoring groundwater contributions in streamflow modelling can lead to significant errors, such as the underestimation of baseflows and the misrepresentation of seasonal variations. Spatiotemporally continuous groundwater measurements are

difficult to obtain, which means not all catchments have groundwater information when developing CAMELS-style datasets. On the other hand, well-developed national-scale hydrological models with three-dimensional groundwater movement are available in Denmark, which has been calibrated with well measurements of groundwater levels. These simulations provide valuable insights for surface hydrological modelling.

CMALE-DK provides consistent data for 3330 catchments, which cover almost the entire land area of Denmark. The dataset

has a median catchment area of 19.61 km$^2$ with a median elevation of 31.94 m. Around 10%, i.e. 304 of these catchments are gauged and contain observed runoff, but the entirety of catchments in this dataset contains consistent simulated hydrological data, including simulated runoff and groundwater dynamics, in addition to hydroclimatic forcing and catchment attributes. These simulations are from a spatially distributed hydrological model, the National Hydrological Model of Denmark (DK-model), which has been thoroughly validated against data from 304 streamflow stations and approximately 40,000 groundwater

wells. The DK-model provides a satisfying and consistent simulation of streamflow and incorporates a sophisticated 3D representation of groundwater processes, including a detailed hydrogeological model. With the first CAMELS datasets to provide a broad range of consistent and high-quality simulated data, we intend to generate a testbed for hybrid or physics-aware ML developments to further accelerate the development of such model types.

The objective of this study is to describe the extensive repository of hydrometeorological time series and catchment attributes

in Denmark. We followed the routine of previous CAMELS dataset and ensure this dataset is comparable and interoperable to the already existing CAMELS datasets. Additionally, some new features are introduced which have not been found in the previously published CAMELS datasets, such as the incorporation of simulation results (shallow/deep groundwater levels,

irrigation) of a spatially distributed hydrological model and features of hydrogeological model and observed groundwater abstraction for irrigation. The dataset follows the FAIR guiding principles for scientific data management and stewardship.

We believe this dataset will enrich the existing CAMELS database with catchments that have different features and hydrological regimes than the already existing datasets, with respect to groundwater influence and size. The paper is organized as follows: Section 2 describes the catchment delineation. Section 3 presents the dynamic variables including climate forcings, observed streamflow, and simulations from the hydrological model. Section 4 presents the sources and features of catchment attributes of topography, soil types, land use, and geology. In Section 5, we discuss the hydrological processes in Denmark

based on the provided dataset. Section 6 contains a short summary of the paper.

## 2 Catchment delineation

Denmark has been divided into 3351 topographical catchments (Højberg et al., 2021), referred to as ID15 catchments (Fig. 1). The division was originally prompted by the need to know the topographic areas upstream monitoring stations, lakes, and marine waterbodies. Thus, enabling the calculation of runoff and loads of nutrients, suspended matter and other chemical

components coupled to the corresponding topographical area. For modelling purposes, the observed catchments were further subdivided and supplemented by a delineation also for unmonitored catchments. A topology for the sub-catchments was established, describing the downstream relations.

The catchment delineation seeks to provide sub catchment areas at a size of about 15 km$^2$ (hence the name ID15 catchments) where possible. However, the criteria that all lake outlets and monitoring stations must coincide with an outlet from a sub-

catchment introduces sub catchments sometimes considerably smaller than 15 km$^2$. Alongside the establishment of new lakes or additional monitoring stations the delineation is subject to periodically updates, with additions of new or alteration of existing catchment boundaries. The most-update version of ID15 catchments includes 3351 sub-catchments with a median area of 12.98 km$^2$, varying from less than one square kilometer to more than 100 km$^2$ for the ID15 catchments containing the largest Danish Lake. Both very large and some very small ID15 catchments were created to accommodate lake catchments and monitoring

stations. The delineation of the ID15 catchments follows topographic catchments. However, especially in flat areas and areas with modified streams and canals, local knowledge of such alterations was accounted for to delineate the actual surface water catchments. Additionally, ID15 catchments also coupled with the National Hydrological Model of Denmark (details are in Section 3.3), time series of simulated streamflow are available at catchment outlets where stream are included in the national hydrological model.

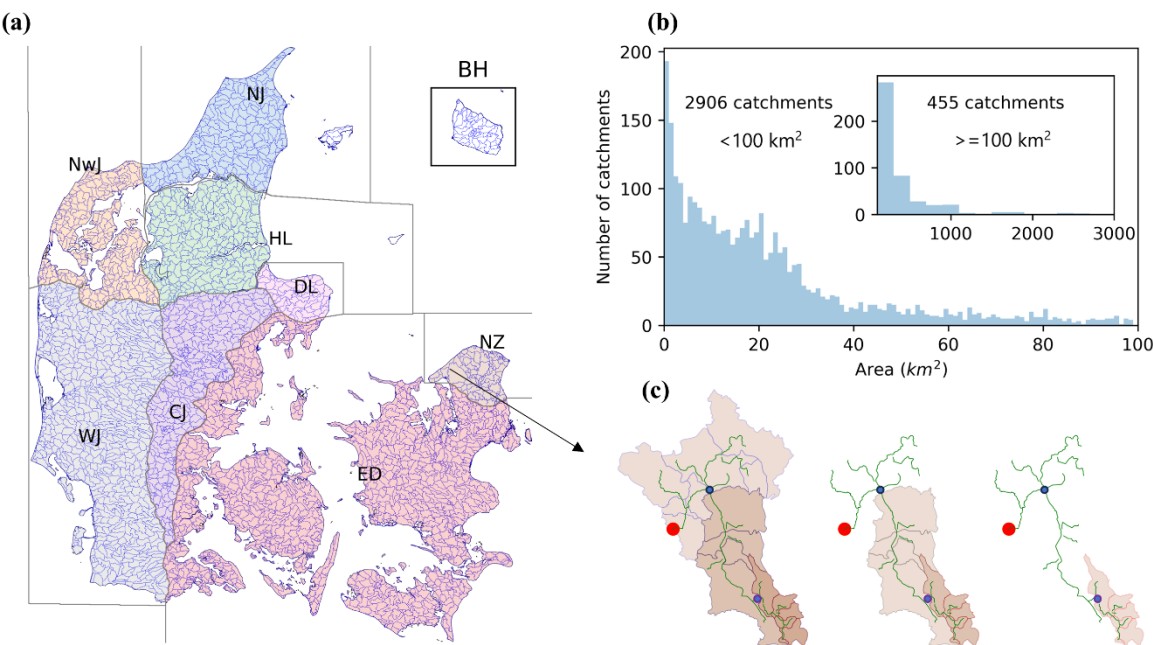

**Figure 1. CAMELS-DK domain. (a) ID15 catchments and geological regions of Denmark, including northern Zealand (NZ), eastern Denmark (ED), western Jutland (WJ), central Jutland (CJ), Djursland (DL), Himmerland (HL), northwestern Jutland (NwJ), Northern Jutland (NJ), and Bornholm (BH), (b) distribution of catchment area, and (c) an example of river channels and catchment boundaries delineated by ID15 catchments in Northern Zealand. The green lines are river channels, red dot indicates the location of a hydrological station downstream the river section, blue dots are examples basin outlets, and the corresponding catchment upstream drainage area are filled with light colour.**

## 3 Dynamic variables

### 3.1 Climate data

The climate time series are based on national gridded products produced by the Danish Meteorological Institute (DMI). Datasets with daily timesteps going back to the year 1989 are based on observations from a network of in-situ weather stations. Daily rainfall time series, aggregated to catchment scale with an area weighted mean function, were derived from the 10 km$^2$ 'Klimagrid Danmark' data from DMI (Scharling, 1999b). The gridded dataset is based on an inverse-distance interpolation of daily observations of precipitation at DMI's stations. The station density changes throughout the years, but generally moves between 200 and 500 stations, distributed across all of Denmark. The impact of the reduction in the number of gauges was evaluated by DMI, and found to be minimal at national scale, but can have local impacts (Andersen, 2021). As many hydrological applications are extremely sensitive to the exact amount of precipitation, the raw observations of precipitation were corrected for sensor/gauge undercatch. Operational precipitation gauges are typically situated 1–1.5 meters above the ground and are influenced by turbulence around the gauge. This turbulence leads to a systematic undercatch of measured precipitation compared to true precipitation. The extent of undercatch depends on wind speed and precipitation type, with particularly significant undercatch observed for snow during windy conditions. However, even liquid precipitation is subject

to undercatch. In Denmark, the undercatch is estimated to average around 5–10% in summer and 15–20% in winter. This systematic bias results in an underestimation of true precipitation, making correction essential for any water resources analysis or hydrological modeling exercise. The precipitation correction model applied in this study is based on a Danish study of the Hellman raingauge. The model uses empirical relationships to estimate undercatch based on wind speed, shelter conditions at the gauge (turbulence), and temperature (to determine precipitation type—liquid, solid, or mixed). Additionally, the model accounts for wetting losses caused by evaporation of water from the inner sides of the rain gauge before measurement. The correction was applied dynamically, accounting for rainfall intensity, wind speed and temperature (solid or liquid precipitation). CAMELS-DK provides corrected precipitation at ID15 catchments scale based on freely available uncorrected precipitation observations, wind speed and temperature data in 10km grid format and the correction approach described by (Stisen et al., 2011). The uncorrected or DMI corrected precipitation data are not part of this dataset. Similarly to the precipitation dataset, DMI also provides gridded datasets of air temperature and potential evapotranspiration (Scharling, 1999a). Potential evapotranspiration is calculated based on the Makkink formula (Van Kraalingen and Stol, 1997) by DMI, specifically modified for Danish conditions. Temperature and potential evapotranspiration are provided at daily timesteps; however, at an original resolution of 20 km$^2$.

The climate data was downloaded directly from the DMI Application Programming Interface (Frie Data). Precipitation correction was then applied to all grids to account for the undercatch biases. The raster data was subsequently clipped to each catchment boundary, and a mean daily value was calculated based on all grids that fall within or touch the boundaries. This process forms the climate time series included in CAMELS-DK.

**Table 1. Dynamic variables in CAMELS-DK. The time series are in daily scale, and spatially aggregated with the mean function.**

| Time series class | Time series name | Description | Unit | Data source |
|---|---|---|---|---|
| Climate variables | precipitation | Daily accumulated precipitation | mm·d$^{-1}$ | Danish Meteorological Institute (Frie Data) |
| | temperature | Daily mean temperature | °C | |
| | pet | Daily potential evapotranspiration (Makkink) | mm·d$^{-1}$ | |
| Hydrological variables | Qobs | Observed streamflow | m$^3$·s$^{-1}$ | (overfladevandsdatabasen ODA, 2020) |
| | Qdkm | Simulated streamflow | m$^3$·s$^{-1}$ | DK-model (Stisen et al., 2020) |
| | DKM_wcr | average water content in root zone | - | |
| | DKM_dtp | phreatic depth to surface layer | m | |
| | DKM_eta | actual evapotranspiration | mm·d$^{-1}$ | |
| | DKM_ rec | Total recharge to saturated zone | mm·d$^{-1}$ | |
| | DKM_sdr | saturated zone drainage flow from point | m$^3$·s$^{-1}$ | |
| | DKM_sre | saturated zone exchange flow with river | m$^3$·s$^{-1}$ | |
| | DKM_gwh | groundwater head in major deeper aquifer layers | m | |

| | DKM_irr | Irrigation | $m^3 \cdot s^{-1}$ | |
| Groundwater abstraction | abstraction | Annual abstractions from groundwater | $m^3 \cdot s^{-1}$ | (National well database) |

## 3.2 Observed streamflow

Streamflow is provided by Aarhus University through the surface water database (overfladevandsdatabasen ODA, 2020).
Water levels are measured sub-daily by hydraulic sensors at hydrological stations and aggregated to daily values. Daily
streamflow is calculated by the sub daily water levels applying these to the established dynamic rating curves. The rating
curves vary throughout the year due to changes in river cross sections caused by erosion and sedimentation or regulation, and
by vegetation growth and -cutting. Mean daily water discharges are calculated from sub-daily time series and are accessible
through overfladevandsdatabasen ODA. Quality control is performed by the Environmental Protection Agency. Streamflow
time series are available for over 1000 hydrological stations spread across Denmark. However, many stations are unsuited for
use in hydrological modelling. Some stations may have limited time series lengths (basin_id 35321223 has the shortest
streamflow record of 1,762 days during the period 1989–2019 in CAMELS-DK), some stations may contain questionable
discharge values. Others may show anthropogenic influence (e.g. owing to the operation of sluices or pumping stations), which
cannot be represented adequately in hydrological models. This meant that, based on experience with the calibration and
validation of the DK-model, the observations dataset was limited to 304 stations (Stisen et al., 2020). Additionally, some of
the stations (4 stations) were located in the middle of a ID15 catchment, these stations were excluded from the dataset. The
spatial distribution of hydrological stations is displayed in Fig 2a. Small catchments (area < 20 km$^2$, 37 stations) are located in
the upstream of larger river systems and in coastal areas. Large catchments (area > 200 km$^2$) are mainly located on the Jutland
peninsular. The largest river of Denmark, River Gudenå, in central Jutland, has a catchment area of 2602 km$^2$ at the most
downstream station.

Data availability for the 304 stations included in CAMELS-DK is illustrated in Fig. 2. Most stations provided continuous
streamflow observations during the earlier part of the study period, specifically from 1989 to 2005 (Fig. 2b). However, the
number of active hydrological stations declined sharply between 2005 and 2010, with only about 200 stations continuing to
provide full observations of streamflow. Fig. 2c provides further insight into data availability based on the fraction of days
with streamflow observations. During 1990–1999, around 280 stations had data for 90% of the period, indicating high data
coverage during this decade. However, data availability declined significantly in subsequent years, with only about 200 stations
retaining 90% data coverage over the entire study period from 1990 to 2019. This decline highlights the challenges of
maintaining consistent long-term hydrological monitoring. The reduction in available stations and data coverage may be due
to factors such as station decommissioning, operational challenges, or changes in monitoring priorities. These trends have
important implications for hydrological modelling, as reduced data availability may limit the spatial and temporal robustness
of hydrological analyses.

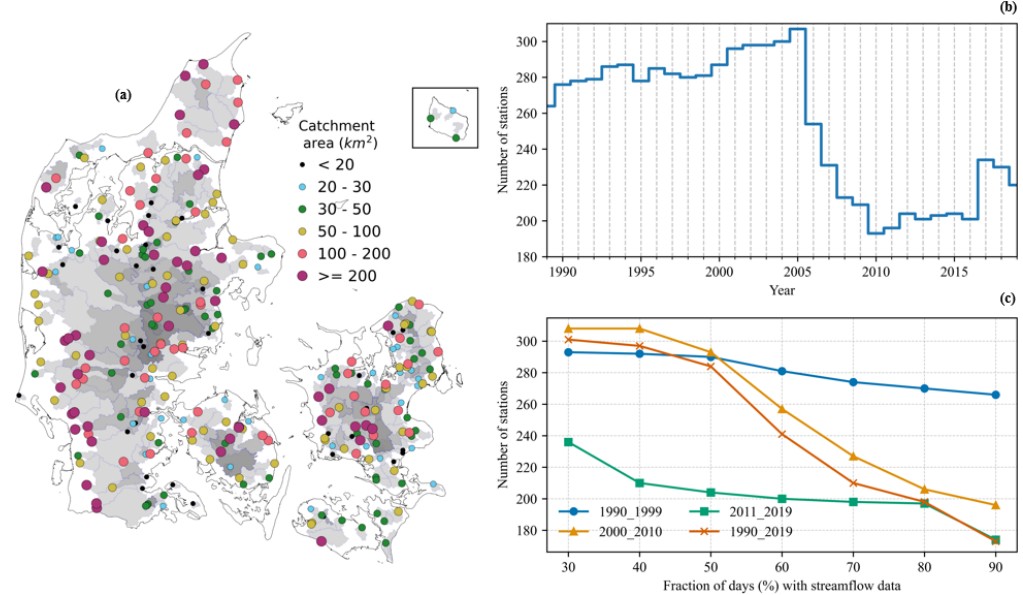

**Figure 2. Streamflow data availability for all gauges contained in the CAMELS-DK dataset. (a) The distribution of hydrometrical stations and adherent catchments, (b) number of stations with full observation time series per year, and (c) the availability of observed streamflow for different periods shown by the number of stations with percentage of available streamflow.**

### 3.3 DK-model simulations

The National Hydrological Model of Denmark (hereafter referred to as DK-model) is an integrated groundwater surface water model covering most of the Danish land area of approximately 43,000 km². It has been continuously developed at the Geologic Survey of Denmark and Greenland (GEUS) since the late 1990's (Henriksen et al., 2019; Højberg et al., 2013; Koch et al., 2021; Schneider et al., 2022; Stisen et al., 2020). It has been used in public consultancy projects, targeting, for example, water resource, evaluation of the impact of climate change on the hydrological cycle, or nitrate retention. However, the DK-model as a whole or in parts also has been and is being used actively in various national and international research projects, further pushing its development (Mahmood et al., 2023; Martinsen et al., 2022; Seidenfaden et al., 2022; Soltani et al., 2021).

The DK-model is setup in the MIKE SHE model code (Abbott et al., 1986; DHI, 2020), which is used to fully couple a 3D description of subsurface flow to 2D overland flow, a simple two-layer description of the unsaturated zone as well as 1D kinematic routing of streamflow. It runs as a transient model at a maximum simulation timestep of 24 hours, and, for the historic period covered by this dataset, with daily climate forcing of precipitation, potential evapotranspiration and temperature (as described in section 3.1). In its most recent version, which serves as a basis for the data presented here, it exists at horizontal resolutions of both 100 m and 500 m (Henriksen et al., 2021),where we are using results from the 500 m version (input data, such as the hydrostratigraphic model, remains at 100 m resolution and is resampled internally by the model). Special attention has been paid to the representation of surface and surface-near processes (shallow groundwater). The model has been calibrated for the period 2000 to 2010 against 304 timeseries of daily streamflow observations (mostly identical to the ones provided in this dataset), and against groundwater head observations from roughly 40,000 intakes distributed across the entire country .

The most significant anthropogenic impacts are accounted for in the model: Groundwater extraction for domestic and industrial water use is included, using data on annual average extractions from waterworks in the national well database Jupiter (National well database, 2024). Groundwater extraction for irrigation in agriculture, which occurs predominantly in the western parts of Denmark, is accounted for based on dynamic calculations of soil water deficit in MIKE SHE (Danapour et al., 2021). Where sewage plants discharge their outflow into streams, these are included as point sources in the stream network, based on yearly averages. As roughly half of the agricultural land in Denmark is artificially drained, saturated zone drainage is included (Møller et al., 2018b). Due to the coarse grid resolution and the related inability to represent the drainage infrastructure explicitly as well as the effects of microtopography, and other aspects such as urban sewer systems acting as further drainage, the DK-model includes drainage across most of Denmark. It is parameterized using drain depths and time constants distributed based on the BASEMAP land use map. Generated drain flow is routed to the river network.

With hydrological processes in Denmark in general, and streamflow in particular, being strongly groundwater-dependent (Duque et al., 2023), we provide, amongst others, groundwater related variables from the DK-model simulations. We could previously show that data-driven methods for streamflow estimation benefit from information contained in simulated groundwater levels across Denmark (Liu et al., 2024). Hence, as part of the CAMELS-DK dataset, we include various simulation results from the DK-model at daily timesteps with relevance for streamflow (Table 1). For reference, we also include the simulated streamflow at the outlet of each ID15 catchment. The other simulation results are aggregated per ID15 catchment. From the two-layer unsaturated zone module of MIKE SHE, the water content in the root zone is provided, together with simulated actual evapotranspiration and recharge from the unsaturated to the saturated zone. For recharge, positive values indicate flow downwards to the saturated zone, whereas negative values indicate upwards flow from the saturated zone, occurring for example in areas of upwelling groundwater. Furthermore, the simulated exchange between streamflow in rivers and the saturated zone gives is provided, as it gives an indication of the groundwater baseflow contribution to streamflow (most Danish rivers can be classified as predominantly gaining streams). Another important part of the hydrological cycle in Denmark is artificial drainage, because roughly half of the Danish agricultural land is artificially drained (Møller et al., 2018a); therefore simulated artificial drain is included in the present dataset. Together, direct groundwater contribution to the streams and artificial drain account for most of the streamflow generated in the DK-model (Refsgaard et al., 2022). Finally, groundwater state is included by two variables: Depth to phreatic representing the depth below the surface to the uppermost simulated groundwater table (which can be a perched groundwater table), as well as the simulated groundwater head in the main deeper aquifer (typically in a few tens of metres depth). The former gives an indication of surface-near interactions between groundwater and surface water, reacting more quickly to precipitation events, whereas the latter typically shows a more delayed response to climate.

### 3.4 Groundwater abstraction

Domestic and industrial water supply in Denmark is almost exclusively covered by groundwater. Data on groundwater abstraction from water works are reported to the national well database Jupiter (Hansen and Pjetursson, 2011), as aggregated

yearly values per well or well field. Those data are used as input to the DK-model and are also provided as part of the CAMELS-DK dataset. Due to the close coupling between surface and subsurface processes in Denmark, groundwater extraction locally impacts streamflow, as for example acknowledged in national water resources assessments (Henriksen et al., 2008, 2023).

## 4 Catchment attributes

### 4.1 Location and topography

The dataset is organized based on the ID15 catchments (Højberg et al., 2021). All the catchments are identified with an eight-digit name according to the shapefile. The locations of catchment outlets are provided as longitude and latitude with the projection of WGS 84 / UTM zone 32N (EPSG:32632). Flow direction indicates the downstream catchments, the column was filled with –99 for most downstream catchments. The number of upstream catchments and the list of upstream catchment IDs are also provided. The area of the individual ID15 catchment and their entire upstream accumulated area originate from the provided shapefile. Gauge types indicate if the catchment is gauged or ungauged, when 'True' indicates that the catchment is a gauged catchment, observed and simulated streamflow are available in the dataset and 'False' indicates only simulated streamflow is available. Details about the observed data, such as the length and the percentage of available observed streamflow data during 1989 to 2019, are provided. Catchment topography described using various statistics derived from elevation and slope are also provided in the attributes table, see Table 2. Please be aware that catchment zonal statistics are based on the entire upstream area.

**Table 2. Catchment location and topography**

| Attribute name | Description | type/Unit | Data source |
|---|---|---|---|
| catch_id | Catchment identifier (eight-digit name of ID15 catchments) | | |
| catch_outlet_lon | Catchment outlet longitude | m | |
| catch_outlet_lat | Catchment outlet latitude | m | Højberg et |
| catch_flow_dir | Catchment flow direction | | al.,(2021) |
| catch_accum_number | The total number of upstream accumulated ID15 catchments | | |
| catch_area | Catchment accumulated area | $m^2$ | |
| gauged_type | Boolean values to indicate the catchment is gauged (True) or ungauged (False) | | |
| gauge_record_pct | If gauged catchment, the time percentage with available observed streamflow in the period from 1989-01-02 to 2019-12-31 | % | |
| elev_min | Minimum elevation of the catchment | m | |
| elev_median | Median elevation of the catchment | m | |

| | | |
|---|---|---|
| elev_mean | Mean elevation of the catchment | m |
| elev_max | Maximum elevation of the catchment | m |
| slope_min | Minimum slope of the catchment | m·km⁻¹ |
| slope_median | Median slope of the catchment | m·km⁻¹ |
| slope_mean | Mean slope of the catchment | m·km⁻¹ |
| slope_max | Maximum slope of the catchment | m·km⁻¹ |
| pct_flat_area | Percentage of catchment area with slope smaller than 3m·km⁻¹ | % |

| Attribute | Description | Unit | Source |
|---|---|---|---|
| elev_mean | Mean elevation of the catchment | m | |
| elev_max | Maximum elevation of the catchment | m | |
| slope_min | Minimum slope of the catchment | $m \cdot km^{-1}$ | Denmark's |
| slope_median | Median slope of the catchment | $m \cdot km^{-1}$ | Height Model - |
| slope_mean | Mean slope of the catchment | $m \cdot km^{-1}$ | Surface |
| slope_max | Maximum slope of the catchment | $m \cdot km^{-1}$ | |
| pct_flat_area | Percentage of catchment area with slope smaller than 3m·km⁻¹ | % | |

300

## 4.2 Climate indices

Climate indices are based on the aforementioned grided datasets of precipitation, temperature, and potential evapotranspiration. To provide consistency with previous CAMELS datasets, we compute the same climatic indices for all catchments in CAMELS-DK as Addor et al. (2017) based on the script provided by Hao et al. ( 2021), such as the mean value of daily

305 precipitation, seasonality, and frequency. Details of these climate indices are listed in Table 3. Climatic indices in CAMELS-DK are calculated based on the time series from 1989 to 2019, which is consistent with the availability of observed discharge.

**Table 3. Climate indices**

| Attribute | Description | Unit |
|---|---|---|
| p_mean | Mean of daily precipitation | $mm \cdot d^{-1}$ |
| t_mean | Mean of daily temperature | °C |
| pet_mean | Mean of daily potential evapotranspiration | $mm \cdot d^{-1}$ |
| aridity | aridity (ratio of mean PET to mean precipitation) | - |
| p_seasonality | seasonality and timing of precipitation (estimated using sine curves to represent the annual temperature and precipitation cycles; positive (negative) values indicate that precipitation peaks in summer (winter); values close to 0 indicate uniform precipitation throughout the year) | - |
| frac_snow | fraction of precipitation falling as snow (i.e., on days colder than 0 °C) | % |
| high_prec_freq | frequency of high precipitation days (≥5 times mean daily precipitation) | $d \cdot yr^{-1}$ |
| high_prec_dur | average duration of high precipitation events (number of consecutive days ≥5 times mean daily precipitation) | d |
| high_prec_timing | season during which most high precipitation days (≥5 times mean daily precipitation) occur | season |
| low_prec_freq | frequency of dry days (< 1 mm·d⁻¹) | $d \cdot yr^{-1}$ |
| low_prec_dur | average duration of dry periods (number of consecutive days < 1 mm·d⁻¹) | d |
| low_prec_timing | season during which most dry days (< 1 mm·d⁻¹) occur | season |

## 4.3 Streamflow signatures

Hydrologic signatures are derived from the observed daily streamflow at 304 hydrological stations and the DK-model-simulated streamflow at the outlets of 2,942 catchments (388 catchments have no simulated river flows) during 1989 to 2019. The signatures are calculated using a MATLAB toolbox called TOSSH (A Toolbox for Streamflow Signatures in Hydrology), developed by Gnann et al. ( 2021). TOSSH provides basic signatures such as magnitude, frequency, duration, timing, and rate of change of a natural streamflow regime, as well as signatures from benchmark papers, including those used by Addor et al. (2017), and an extended set of process-based signatures.

We calculated 13 hydrological signatures presented by Addor et al. (2017) using TOSSH, ensuring consistency with other CAMELS datasets (e.g., Alvarez-Garreton et al., 2018; Chagas et al., 2020; Höge et al., 2023). These signatures are listed in Table 4. Additionally, we provide process-based signatures calculated with TOSSH, including groundwater and overland flow signatures defined by McMillan (2020), see Appendix A. The groundwater signatures characterize groundwater storage, groundwater dynamics, and baseflow, while the overland flow signatures reflect infiltration excess and saturation excess in overland flow.

**Table 4. Hydrographical signatures based on observed and simulated streamflow.**

| Attribute | Description | Unit |
|---|---|---|
| q_mean | mean daily streamflow | mm·d$^{-1}$ |
| TotalRR | total runoff ratio (runoff divided by the precipitation) | |
| QP_elasticity | streamflow precipitation elasticity (sensitivity of streamflow to changes in precipitation at the annual timescale, using the mean daily streamflow as reference) | |
| FDC_slope | slope of the flow duration curve (between the log- transformed 33rd and 66th streamflow percentiles) | |
| BFI | baseflow index (ratio of mean daily baseflow to mean daily streamflow, hydrograph separation performed using the Ladson et al., 2013 digital filter | |
| HFD_mean | mean half-flow date (date on which the cumulative streamflow since October reaches half of the annual streamflow) | day of year |
| Q5 | 5% flow quantile (low flow | mm·d$^{-1}$ |
| Q95 | 95% flow quantile (high flow | mm·d$^{-1}$ |
| high_Q_frequency | frequency of high-flow days (> 9 times the median daily flow | |
| high_Q_duration | average duration of high-flow events (number of con- secutive days > 9 times the median daily flow | d |
| low_Q_frequency | frequency of low-flow days (< 0.2 times the mean daily flow | |
| low_Q_duration | average duration of low-flow events (number of consec- utive days < 0.2 times the mean daily flow | d |
| zero_Q_frequency | frequency of days with Q=0 | |

## 4.4 Land use and land cover

We provide both national and continental data sources for land use and land cover (LULC). The LULC attributes are derived from two datasets: the CORINE Land Cover (CLC) from the European Copernicus program (Büttner et al., 2004) and Basemap04 developed by Aarhus University (Levin, 2022). CAMELS-DK calculates the catchment-scale percentage of key

LULC classes. The aggregated areal percentages of five land cover types are provided for all catchments, including the percentages of forest, crops, water, urban areas, and wetlands (Table 5).

The CORINE Land Cover (CLC) product offers a pan-European land cover and land use inventory with 44 thematic classes, ranging from broad forested areas to individual vineyards (Büttner et al., 2004). The product is updated every six years with new status and change layers for 1990, 2000, 2006, 2012, and 2018. The dataset is based on multiple satellite data sources,

such as Landsat, SPOT-4/5, RapidEye, and Sentinel-2. The geometric accuracy of the CLC data is 100 meters for the first dataset released in 1990, with accuracy improving to better than 100 meters in subsequent years. Basemap04 provides four maps of LULC representing the years 2011, 2016, 2018, and 2021, with a spatial resolution of 10 meters. The dataset combines existing thematic geographic information, such as census mapping of state forests, maps of protected habitats, agricultural field parcel maps, and cadastre maps (Levin, 2022).

**Table 5. Land use attributes.**

| Attribute | Description | Years | Unit | Data source |
|---|---|---|---|---|
| pct_ forest_corine_yyyy | Percentage of agricultural area | | % | |
| pct_ crop_corine_yyyy | Percentage of grass and herb vegetation area | | % | |
| pct_ water_corine_yyyy | Percentage of medium-scale vegetation area | 1990, 2000, 2006, 2012, 2018 | % | CORINE Land Cover (2024) |
| pct_ urban_corine_yyyy | Percentage of deciduous forest area | | % | |
| pct_ wetlands_corine_yyyy | Percentage of mixed forest area | | % | |
| pct_forest_levin_yyyy | Percentage of forest area | | % | |
| pct_ crop_levin_yyyy | Percentage of agricultural area | | % | |
| pct_ water_levin_yyyy | Percentage of lake and stream area | | % | |
| pct_ urban_levin_yyyy | Percentage of urban area | 2011, 2016, 2018, 2021 | % | Levin (2022) |
| pct_ naturewet_levin_yyyy | Percentage of natural area (wet, agriculture, extensive) | | % | |
| pct_ naturedry_levin_yyyy | Percentage of natural area (dry, agriculture, extensive) | | % | |

## 4.5 Soil types

Physical soil properties were derived from national and continental data sources. For the former, catchment average soil texture

was derived for four soil layers, i.e., 0–5, 5–15, 15–30, 30–60, 60–100, and 100–200 cm at ~30 m spatial resolution. The Danish dataset by Adhikari et al. (2013) is based on a regression model using field data for 1958 soil profiles and 17 environmental covariates. In addition, catchment average soil physical properties were derived from the European Soil Data Centre (ESDAC) and the dataset of 3D Soil Hydraulic Database of Europe at 1 km and 250 m resolution (Hiederer, 2013b, a; Tóth et

al., 2017). A broader set of variables was processed, including rooting depth, saturated hydraulic conductivity, among others.

Soil maps of ESDAC are at 1000 m spatial resolution and are based on data from the European Soil Database in combination with data from the Harmonized World Soil Database (HWSD) and Soil-Terrain Database (SOTER). Several hydraulic parameters from the 3D Soil Hydraulic Database of Europe, available at 1 km and 250 m resolutions, are included in CAMELS-DK, such as water content at field capacity and saturated hydraulic conductivity (Table 6).

**Table 6. Catchment attributes of soil types.**

| Attribute | Description | Unit | Data source |
|---|---|---|---|
| pct_sand | Sand content | % | |
| pct_silt | Silt content | % | |
| pct_clay | Clay content | % | |
| pct_organic | Organic carbon content | % | Hiederer (2013a, b) |
| bulk_density | Bulk density | $g \cdot cm^{-3}$ | |
| tawc | Total available water content (from PTR) | mm | |
| pct_gravel | Coarse Fragments | % | |
| root_depth | depth available for roots | m | |
| FC | water content at field capacity | $cm^3 \cdot cm^{-3}$ | |
| HCC | hydraulic conductivity curve | $\log_{10} [cm \cdot d^{-1}]$ | |
| KS | saturated hydraulic conductivity | $\log_{10}[cm \cdot d^{-1}]$ | Tóth et al. (2017) |
| MRC | moisture retention curve | $cm^3 \cdot cm^{-3}$ | |
| THS | Saturated water content | $cm^3 \cdot cm^{-3}$ | |
| WP | water content at wilting point | $cm^3 \cdot cm^{-3}$ | |
| pct_claynor_30 | Clay percentage in layer depth of 0-30cm | % | |
| pct_ claynor_60 | Clay percentage in layer depth of 30-60cm | % | |
| pct_ claynor_100 | Clay percentage in layer depth of 60-100cm | % | |
| pct_ claynor_200 | Clay percentage in layer depth of 100-200cm | % | |
| pct_ fsandno_30 | Fine sand percentage in layer depth of 0-30cm | % | |
| pct_ fsandno_60 | Fine sand percentage in layer depth of 30-60cm | % | |
| pct_ fsandno_100 | Fine sand percentage in layer depth of 60-100cm | % | Adhikari et al.(2013) |
| pct_ fsandno_200 | Fine sand percentage in layer depth of 100-200cm | % | |
| pct_ gsandno_30 | Coarse sand percentage in layer depth of 0-30cm | % | |
| pct_ gsandno_60 | Coarse sand percentage in layer depth of 30-60cm | % | |
| pct_ gsandno_100 | Coarse sand percentage in layer depth of 60-100cm | % | |
| pct_ gsandno_200 | Coarse sand percentage in layer depth of 100-200cm | % | |

## 4.6 Hydrogeology and geology

Hydrogeological features were obtained from the 3D hydrogeological model implemented in the DK-Model (Stisen 2019). The model is based on a layer structure, with in total of up to 28 hydrostratigraphical layers that have been modelled based on borehole data and subsurface geophysics. The layers delineate the occurrence of, i.e., quaternary, pre-quaternary sand and clay as well as limestone. The bottom and top of the hydrostratigraphical layers are mapped on a 100 m grid and five key variables were compiled for the CAMELS-DK dataset: (1) Depth to chalk specifies the vertical distance from terrain to chalk. (2) Thickness of the uppermost aquifer accumulates quaternary sand layers that are not intermittent by clay layers thicker than 1 m. (3) Depth to uppermost aquifer specifies the vertical distance from terrain to the top of the uppermost quaternary sand layer thicker than 1 m. 4) Thickness of uppermost clay specifies the accumulated quaternary clay layers from terrain until a sand layer thicker than 1 m is present. 5) Thickness of uppermost sand specifies the accumulated quaternary sand layers from terrain until a clay layer thicker than 1 m is present. All variables are stated in meters and the catchment average has been calculated for the CAMELS-DK dataset (Table 7). The hydrostratigraphical model for the island Bornholm is developed separately from the rest of Denmark. Bornholm's geology is significantly different with Pre-Cambrian granite-gneisses and Cambrian sandstones. Available data are interpreted to build a 3D voxel model for the island. Due to these differences, the five hydrogeological features are not available for Bornholm. Moreover, the 14 classes of the surface geology map of Denmark (Pedersen et al., 2011a) were processed as catchment percentages.

**Table 7. Catchment attributes of hydrogeologic and geologic features.**

| Attribute | Description | Unit | Data source |
|---|---|---|---|
| chalk_d | Depth to chalk | m | |
| uaquifer_t | Thickness of uppermost aquifer | m | Stisen et al. (2020) |
| uaquifer_d | Depth to uppermost aquifer | m | and Ondracek |
| uclay_t | Thickness of uppermost clay | m | (2023) |
| usand_t | Thickness of uppermost sand | m | |
| pct_aeolain_sand | Percentage of aeolian sand, including dunes and cover sands | % | |
| pct_water_deposit | Percentage of freshwater deposits | % | |
| pct_marsh | Percentage of marsh, alternating tin | % | |
| pct_marine_sand | Percentage of Marine sand and clay | % | |
| pct_beach | Percentage of Beach ridges | % | Pedersen et al. (2011a, |
| pct_sandy_till | Percentage of Sandy and gravelly till | % | b) |
| pct_till | Percentage of till, clay and fine-sandy | % | |
| pct_glaf_sand | Percentage of glaciofluvial sand and gravel | % | |
| pct_glal_clay | Percentage of glaciolacustrine laminated clay, silt and fine sand | % | |
| pct_down_sand | Percentage of downwash sandy deposits | % | |

| pct_glam_clay | Percentage of glaciomarine clay, silt and sand | % |
|---|---|---|

## 5 Data Discussion

### 5.1 Regional variability in catchment attributes

Denmark has a gentle topography with many regions at a mean elevation lower than 50 m and low slopes (Fig. 3a, b). Across central Jutland, a ridge is located with elevations reaching up to about 170 m. This ridge, together with the prevailing westerly winds, is responsible for the regional precipitation patterns with higher precipitation in western Jutland, and lower precipitation in the more eastern parts of the country (Fig. 3c). Another important factor controlling hydrology across Denmark is the soil type: soils are most clayey in eastern Denmark, central and northwestern Jutland (Fig. 3d), and least clayey (mostly sandy) in western Jutland. Higher percentage of clay soil generally relates to lower hydraulic conductivities in the topsoil, leading to higher groundwater levels and more artificial drainage. It also leads to larger amount of water available for evapotranspiration and lower runoff. In contrast, the thickness of uppermost sand is high in West Jutland, north Jutland, and Himmerland (Fig. 3e). The depth to chalk aquifers is high (> 250 m) in northwestern Jutland, central and western Jutland, but lower in North Zealand and Djursland with typical depths of around 60 m (Fig. 3f). The temperate climate with precipitation clearly exceeding evapotranspiration, low-lying topography, and geology dominated by clay till or sandy meltwater deposits result in high groundwater levels and abundant groundwater resources in Denmark.

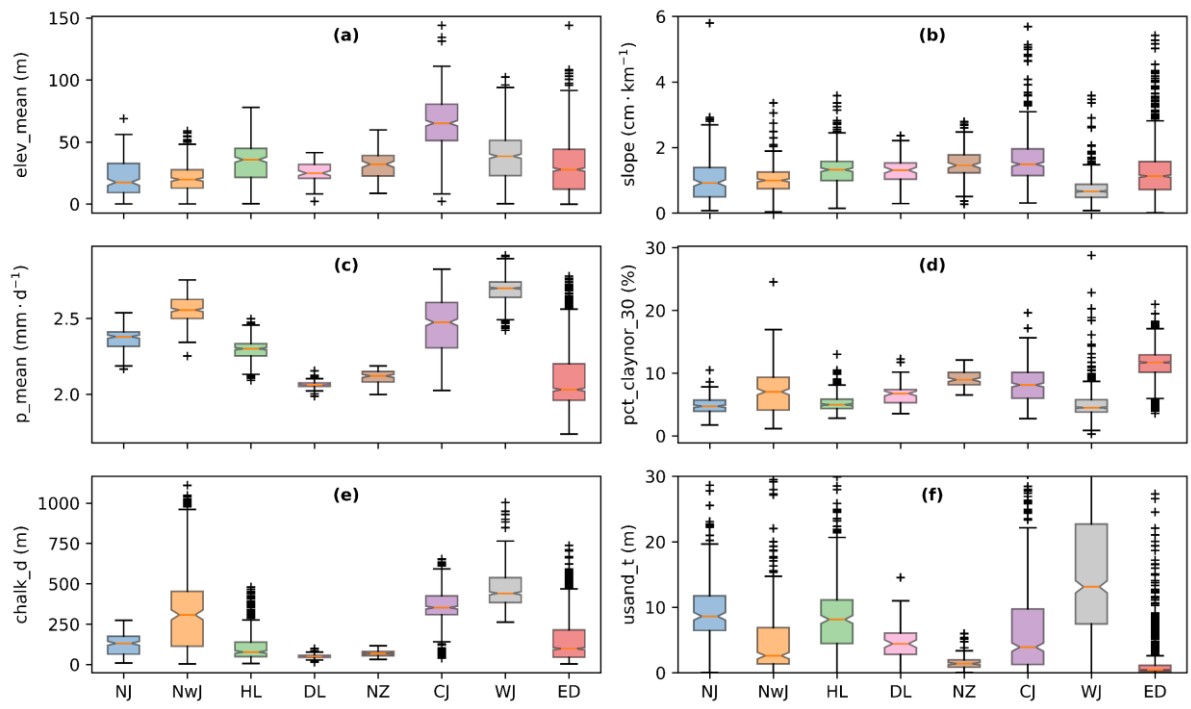

**Figure 3. Regional distributions of (a) mean elevation, (b) slope, (c) mean daily precipitation, (d) Clay percentage in layer depth of 0-30cm, (e) depth of chalk, and (f)** Thickness of uppermost sand in geo-regions (NJ: North Jutland, NwJ: northwestern Jutland, HL: Himmerland, DL: Djursland, NZ: North Zealand, CJ: Central Jutland, WJ: Western Jutland, and ED: East Denmark).

## 5.2 Surface water dynamics

CAMELS-DK provides observed streamflow data from 304 hydrological stations and simulated streamflow data for 2,942 catchments. However, 388 catchments lack observed or simulated streamflow data because they are located upstream or in coastal areas without distinct land surface river channels. The simulated streamflow is based on the DK-model, which was calibrated using observed streamflow data from the period 2000-2010, see section 3.3. Fig. 4 shows the Sutcliffe and Kling-Gupta efficiency (KGE) and Nash–Sutcliffe model efficiency coefficient (NSE) for the DK-model's simulated hydrography, referenced against observations from 1990-1999. The mean NSE is 0.46, and the median value is 0.64. The mean KGE is 0.64, and the median value is 0.69, indicating that the simulated streamflow is satisfactory at many stations, with the simulations and observations showing temporal coherence and consistency.

The DK-Model performs well at the national scale but has challenges in central and northern regions. These challenges arise from factors such as geological complexity, insufficient regional parameterization, and limitations in data quality and station coverage over time. Simulations show reduced performance under high-flow and low-flow conditions, as evidenced by lower NSE values in northern Zealand and central and northern Jutland (Fig. 4). The deteriorated performance in regions like central and northern Jutland is attributed to significant geological variability and limited experience with newer parameterization approaches. Smaller stations also tend to exhibit lower performance compared to larger ones, likely due to their sensitivity to

geological complexity and inadequate parameter regionalization for drainage flows. More details on the DK-Model's performance, including its strengths and limitations, can be found in Stisen et al. (2020). To address these limitations, recent advancements such as deep learning methods and hybrid modelling schemes have been explored to enhance streamflow forecasting (Liu et al., 2024). Furthermore, the DK-Model's performance in ungauged catchments remains insufficiently validated, largely due to the computational demands of model calibration.


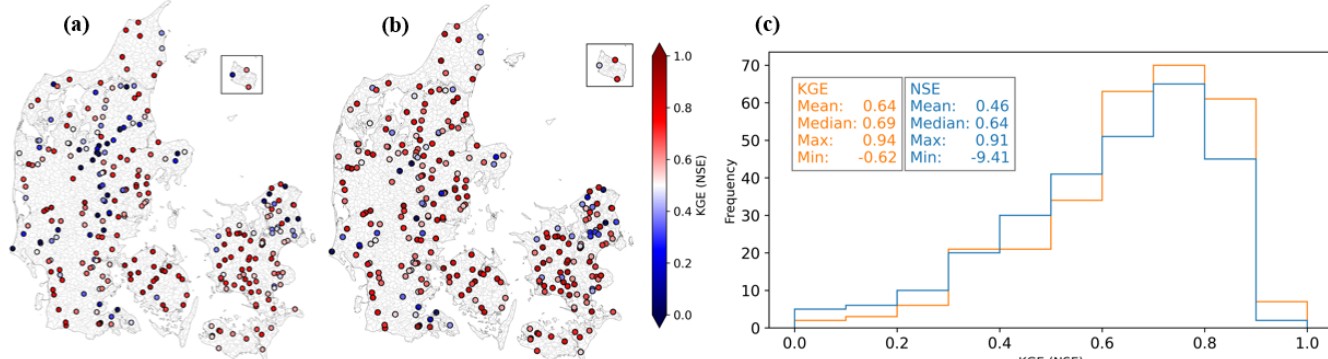

**Figure 4. The performance of DK-model simulated streamflow at 304 hydrological stations. (a) spatial distribution of Nash–Sutcliffe model efficiency coefficient, (b) spatial distribution of the Kling–Gupta efficiency (KGE), and (c) histogram of NSE and KGE.**


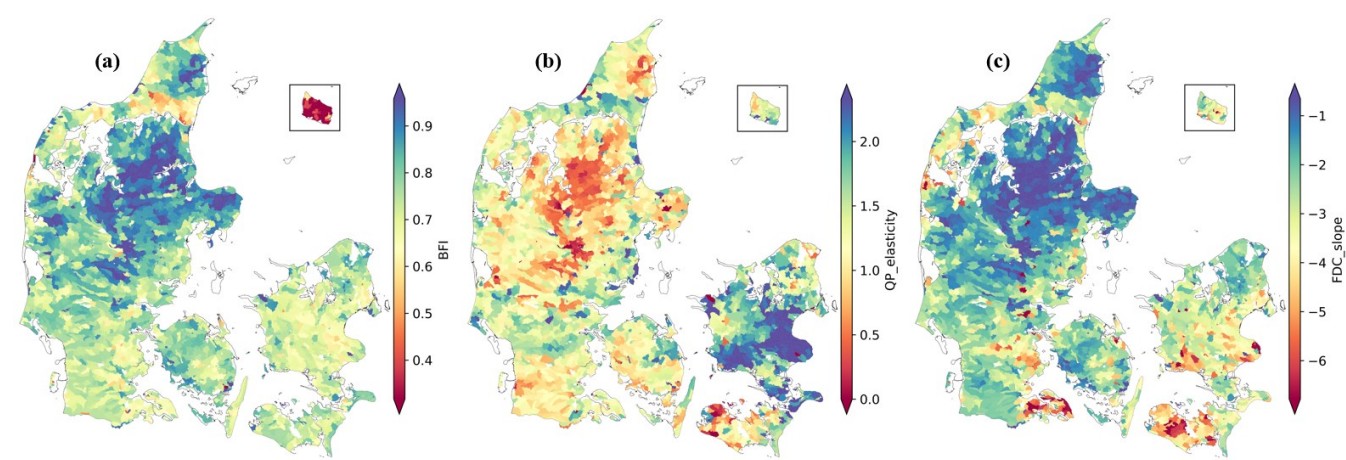

**Figure 5. Spatial distribution of (a) Baseflow index, (b) Streamflow-precipitation elasticity, and (c) Slope of the flow duration curve.**

Groundwater and surface water are closely linked in Denmark (Duque et al., 2023). Fig. 5 presents three hydrographic
signatures derived from DK-model simulations: Baseflow Index (BFI), Streamflow-Precipitation Elasticity (QP-elasticity), and the Slope of the Flow Duration Curve (FDC_slope). BFI represents the proportion of streamflow that occurs as baseflow.

The value is higher in Himmerland, Djursland, and central Jutland, indicating that a significant portion of the streamflow comes from groundwater seepage (Fig. 5a). The groundwater levels are lower in these areas, resulting in the slow and steady release of groundwater into the streams, thereby exhibiting more stable flow conditions throughout the year. Groundwater
buffers against short-term variations in precipitation in these areas. QP-elasticity indicates the sensitivity of streamflow to changes in precipitation, which is low in central Jutland and high in eastern Denmark. This suggests significant regulation of streamflow by groundwater. The FDC_slope also shows higher values in eastern Denmark, indicating a highly variable stream flow that is largely due to the quick runoff of rainfall into the stream (Fig. 5c).

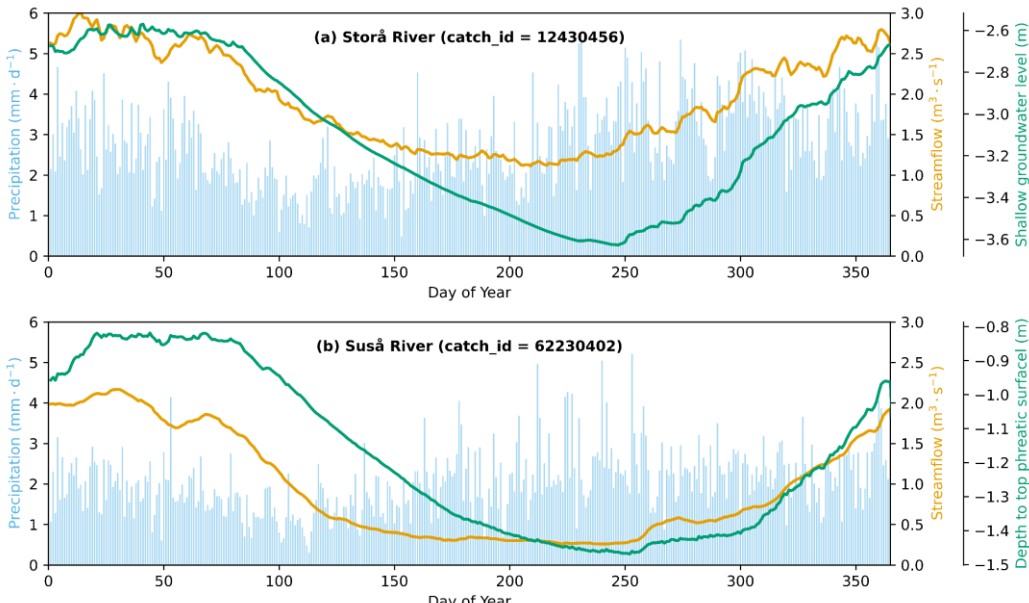

**Figure 6. Climatology of precipitation, observed streamflow, and DK-model simulated shallow groundwater levels in (a) Storå River (the largest river in West Jutland), and (b) Suså River (the largest in East Denmark).**

Groundwater – surface water interaction is significant at temporal scales as well. Fig. 6 shows the climatology (1990 - 2019) of precipitation, observed streamflow and depth to phreatic layers in Storå River (largest river in Western Jutland) and Suså River (largest river in East Denmark), close to each of their outlets. Climatology is calculated by averaging time series values
for the same day of the year across 1990-2019. The spatial variability of precipitation is relatively small, and the seasonality is similar across Denmark. The two examples shown in Fig.6 have a comparable amplitude of precipitation during the year with highest precipitation occurring during the fall and winter months, and lowest precipitation during spring. Streamflow, though, has a more pronounced seasonality, due to higher evapotranspiration during summer, leading to lower runoff coefficients. Even in the climatology, the peaky streamflow response of the sandier Storå catchment becomes apparent,
compared to the more clayey Suså catchment. Moreover, summer baseflow in the Storå catchment is higher than in the Suså catchment (see also BFI in Fig. 5a, and clay percentage in Fig. 3d), which again is due to the better connectivity between (shallow and deep) groundwater and rivers due to the more sandy/higher conductivity subsurface conditions in large parts of

Western Jutland. Linked to similar processes, differences in groundwater dynamics can be seen, with the Storå catchment exhibiting lower groundwater tables, but higher seasonal amplitudes of the two example catchments. The steady decline of groundwater levels during the summer months is due to elevated levels of evapotranspiration, limiting groundwater recharge during that time. Groundwater recharge typically starts again around the beginning of September, and streamflow levels follow closely and begin to rise from summer baseflow conditions around the same time.

**5.3 Groundwater dynamics**

Groundwater supplies nearly 100 percent of Danish domestic and industrial water use. Groundwater monitoring and modelling are thus important for sustainable water management in Denmark. Fig.7 shows catchment-aggregated variables from DK-model simulations, such as depth to phreatic layers (Fig. 7a), groundwater extraction for irrigation (Fig. 7b), and reported groundwater abstractions (Fig. 7c) from waterworks. The spatial pattern of phreatic depth shows the groundwater level is highest in eastern and southern Denmark, and lowest in northern Denmark. In some catchments located in Himmerland and central Jutland, the average phreatic depth is around 15 meters below land surface (Fig. 7a). An example of the time series of phreatic depth in a catchment in Himmerland shows an annual amplitude of approximately 5 meters. The groundwater table is higher in winter and lower in summer, which aligns with the seasonal variations in precipitation and evapotranspiration. Precipitation deficits can reduce the amplitude of phreatic depth, especially during winter months such as in 1996. Catchments with more shallow groundwater depths show typically smaller yearly amplitudes. Groundwater extraction for domestic and industrial water supply is based on yearly data from the national well database Jupiter (National well database, 2024). Groundwater abstraction for irrigation is only available in Jutland since there is no significant irrigation activity in the other parts of Denmark. Irrigation is provided as determined dynamically, demand-based by the DK-model.

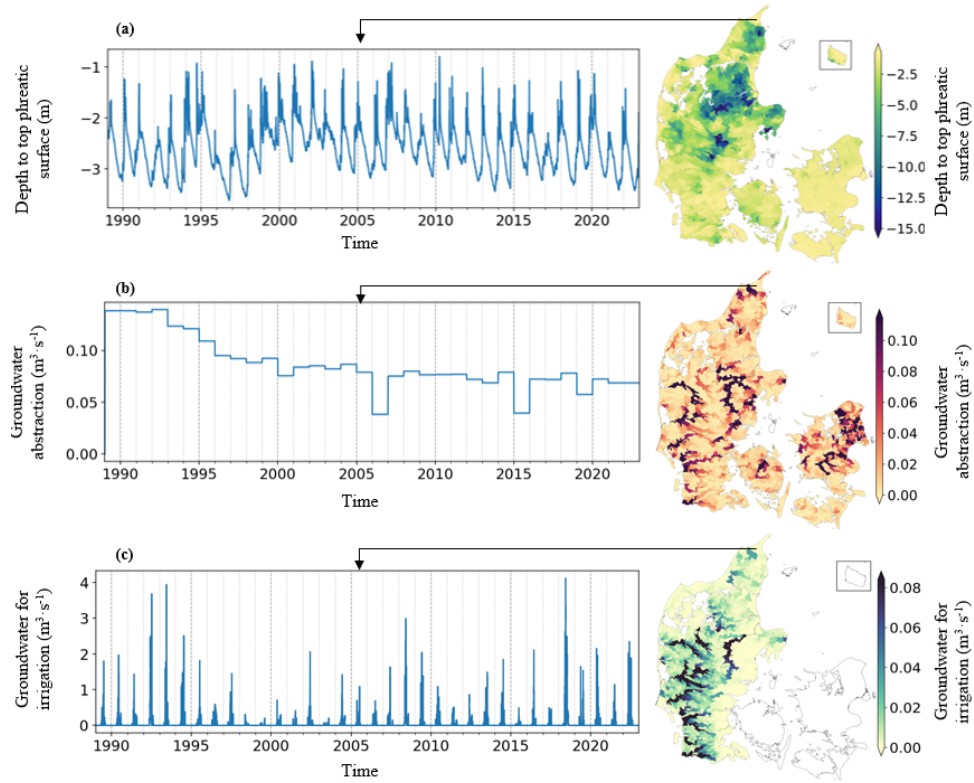

**Figure 7. Groundwater dynamics, abstraction, and groundwater abstraction for irrigation based on CAMELS-DK. The left column illustrates example time series data from a catchment in North Jutland (catch_id = 37220013), while the right column presents a map of time-averaged values for all basins. From top to bottom, the rows represent: (a) depth to the phreatic surface, (b) catchment-wide groundwater abstraction with corresponding annual time series, and (c) catchment-wide groundwater abstraction specifically for irrigation purposes.**

## 6 Dataset structure

The CAMELS-DK dataset is available at https://doi.org/10.22008/FK2/AZXSYP (Koch et al., 2024), data description in file 'Data_description.pdf' details the structure of the dataset. There are four folders after download and unzipping the dataset. Landscape attributes including climate indices, topography, hydrological signatures, land use, soil characteristics, and hydrogeological features are saved in a folder named 'Attributes'. These static attributes are saved in CSV files, where the index is the 8-digital identifier, and the columns are the names of variables (see Tables 2-5). Timeseries of climate data, observed streamflow, DK-model simulated variables, such as depth to phreatic layers, average soil water content, and actual evapotranspiration, and observed yearly groundwater abstractions are available in the folder 'Dynamics'. The timeseries data is saved separately for gauged and ungauged catchments. The shapefiles of catchment boundaries, location and the geo-regions of Denmark are provided in a folder named 'Shapefile'. The Python script of processing the time series and landscape attributes based on original datasets are provided in a folder named 'Python'.

# 7 Conclusion

We present an updated version of CAMELS-DK, which includes extensive hydrometeorological variables and landscape features for 3330 catchments (304 gauged catchments) covering the entirety of Denmark. This dataset is designed to support large-scale, data-driven studies, aligning with the research topics addressed by other CAMELS-style and LSH datasets. Its development follows established conventions, including daily time series of hydrometeorological variables spanning 1989 to 2023, as well as catchment landscape attributes such as climate, soil type, land use/cover, and geology. This ensures seamless integration with other CAMELS-style datasets for large-sample studies.

What sets CAMELS-DK apart is the inclusion of observed annual groundwater abstraction and simulated variables from a process-based hydrological model (PBM)—the National Hydrological Model of Denmark—including simulated discharge, soil water content, and groundwater head. These additional dynamic variables, particularly the groundwater-relevant variables, provide valuable insights for surface water modelling, despite being based on simulations, as they reflect the close interactions between groundwater and surface water in many Danish catchments. We anticipate that this addition of PBM simulations will not only enhance data-driven model performance for surface water modelling in Denmark, but also benchmark and facilitate the development of physics-informed ML algorithms, transfer learning approaches, and hybrid modelling frameworks. We provide detailed instructions on how to access the source data, as we are not permitted to share it directly with CAMELS-DK. For users who wish to calculate catchment-aggregated variables, Python scripts are included for data processing, which can be used to derive spatially aggregated time series and attributes for 3330 catchments.

## Appendix

### Appendix A: Additional hydrological signatures

| Attribute | Description | Unit | Reference |
|---|---|---|---|
| SnowDayRatio | Snow day ratio | – | Euser et al., (2013) |
| RLD | Rising limb density | $1 \cdot d^{-1}$ | |
| AC1 | Lag-1 autocorrelation | – | McMillan, (2020) for groundwater |
| RR_Seasonality | Runoff ratio seasonality | – | |
| EventRR | Event runoff ratio | – | |
| StorageFraction | Ratio between active and total storage | – | |
| Recession_a_Seasonality | Seasonal variations in recession parameters | – | |
| AverageStorage | Average storage from average baseflow and storage-discharge relationship | – | |
| Spearmans_rho | Non-uniqueness in the storage-discharge relationship | – | |
| EventRR_TotalRR_ratio | Ratio between event and total runoff ratio | – | |
| VariabilityIndex | Variability index of flow | – | |
| IE_effect | Infiltration excess importance | – | McMillan, (2020) for surface water |
| IE_thresh | Infiltration excess threshold location (in a plot of quickflow volume vs. maximum intensity) | $mm \cdot d^{-1}$ | |
| IE_thresh_signif | Infiltration excess threshold significance (in a plot of quickflow volume vs. maximum intensity) | – | |
| SE_effect | Saturation excess importance | – | |
| SE_thresh_signif | Saturation excess threshold significance (in a plot of quickflow volume vs. total precipitation) | – | |
| SE_thresh | Saturation excess threshold location (in a plot of quickflow volume vs. total precipitation) | mm | |
| SE_slope | Saturation excess threshold above-threshold slope (in a plot of quickflow volume vs. total precipitation) | – | |

| | | | |
|---|---|---|---|
| Storage_thresh | Storage/saturation excess threshold location (in a plot of quickflow volume vs. antecedent precipitation index + total precipitation) | mm | |
| Storage_thresh_signif | Storage/saturation excess threshold significance (in a plot of quickflow volume vs. antecedent precipitation index + total precipitation) | – | |
| BaseflowMagnitude | Difference between maximum and minimum of annual baseflow regime | mm | Gnann et al., (2021) |
| ResponseTime | Catchment response time | d | |
| FlashinessIndex | Richards-Baker flashiness idex | – | |
| PQ_Curve | Slopes and breakpoints in cumulative P-Q regime curve | – | |
| Q_n_day_max | n-day maximum streamflow | $mm \cdot d^{-1}$ | |
| Q_skew | Skewness of streamflow | $mm^3 \cdot d^{-3}$ | |
| Q_var | Variance of streamflow | $mm^2 \cdot d^{-2}$ | |
| RecessionK_part | Recession constant of early/late (exponential) recessions | $1 \cdot d^{-1}$ | |
| SeasonalTranslation | Amplitude ratio and phase shift between seasonal forcing and flow cycles | – | |
| SnowStorage | Snow storage derived from cumulative P-Q regime curve | mm | |
| zero_Q_duration | Zero flow duration | d | |
| Q_7_day_min | 7-day minimum streamflow | $mm \cdot d^{-1}$ | |
| CoV | Coefficient of variation | – | |
| HFI_mean | Half flow interval | d | |
| BaseflowRecessionK | Exponential recession constant | $1 \cdot d^{-1}$ | |


**Appendix B. Comparison of simulated and observed signatures.**

The DK-model has been calibrated jointly against observed streamflow at around 300 hydrological stations; the performance of DK-model simulated streamflow is satisfactory at many stations (Stisen et al., 2020). Here, we displayed some comparison results of signatures derived from DK-model simulated streamflow and the observations. The magnitude of streamflow, such 510 as the mean streamflow, between simulated streamflow is aligned with the observations (Fig. B1a). The mean streamflow is lower than 5 $m^3 \cdot s^{-1}$ at many stations (96%) and 18 stations are in large rivers with a daily average streamflow higher than 5 $m^3 \cdot s^{-1}$. DK-model tends to overestimate the BFI (Fig.B1b), indicating a slightly smoother streamflow hydrograph simulated by the DK-model. The precipitation elasticity of streamflow, which indicates the sensitivity of streamflow to precipitation, is still challenged to be captured accurately by the model (Fig. B1c). The slope of flow duration curves (FDC_slope) quantifies 515 the variability of hydrographs. A steep slope indicates a highly variable stream, where flow is primarily driven by the quick runoff of rainfall to the stream. The simulated hydrography tends to overestimate the FDC_slope for highly variable streams but underestimate it for less variable streams (Fig. B1d).

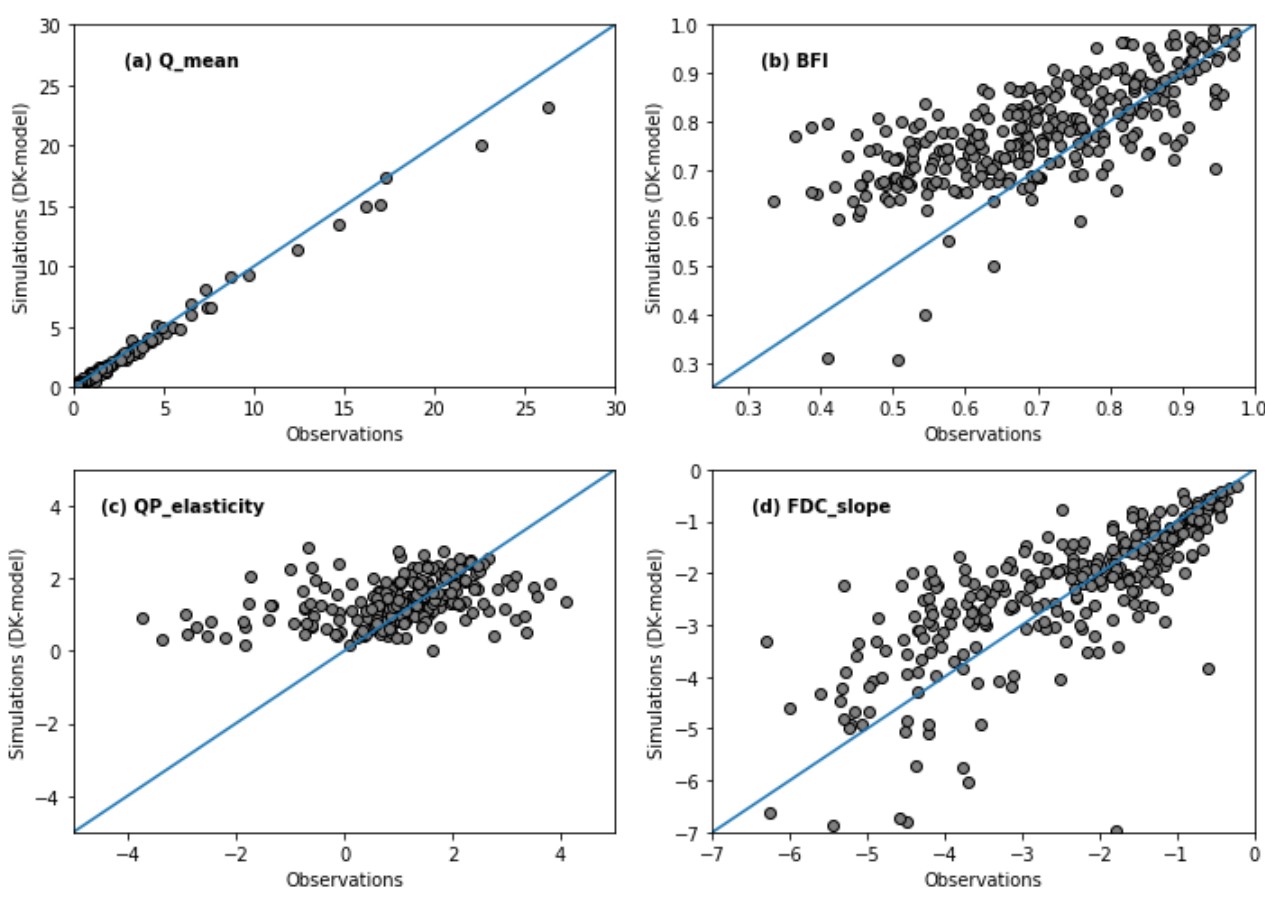

**Figure B1. Comparison of observed signatures and DK-model simulated signatures.**


**Data availability**

The CAMELS-DK dataset is available at the GEUS Dataverse under a Creative Commons Zero license at https://doi.org/10.22008/FK2/AZXSYP (Koch et al., 2024). Since all data is continuously updated by the respective providers, we list the respective sources of their free availability: ID15 catchment shapefile is updated. The newest version of the shapefile

can be requested by writing to id15@ecos.au.dk. High resolution DEM data is available here https://dataforsyningen.dk/data/928. The gridded climate data is created by the Danish Meteorological Institute (DMI) and downloaded through DMI Data API (https://opendatadocs.dmi.govcloud.dk/DMIOpenData). The observed streamflow data can be downloaded from the overfladevandsdatabasen ODA (https://odaforalle.au.dk/). Simulation results of surface water and groundwater dynamics from the National Hydrological model of Denmark are available at

https://dennationalehydrologiskemodel.dk/.

**Competing interests**

The authors declare that they have no conflict of interest.

## Author contribution

JK, RJMS, and JL conceptualized and formulated the overarching research ideas and collected data from different sources. JL processed the data, conducted the formal analysis, and prepared the original draft under the supervision of RJMS and JK. MFTH assisted with data processing. LT, ALH, and HT provided the ID15 catchments, validated the data sources, and reviewed the manuscript.

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
