# Peer review of "CAMELS-DK: Hydrometeorological Time Series and Landscape Attributes for 3330 Danish Catchments with Streamflow Observations from 304 Gauged Stations"

_Earth System Science Data, 2024_

## Referee Comment (RC1)

**General comments**

Thank you for your work compiling the CAMELS-DK dataset and the manuscript. The introduction gives an excellent overview of the current state of large-sample hydrological datasets and the challenges of preprocessing and retrieving large amounts of harmonized hydro-meteorological data, which CAMELS datasets help to solve. The data discussion section gives a very good and helpful overview on the hydrologic characteristics and specificities of Denmark, which is especially helpful for someone who is not familiar with the hydrological landscape in Denmark and can provide great insights and ideas for the discussion of one's own study results.

CAMELS-DK is another great addition to the ever-growing family of CAMELS datasets, especially with the incorporation of (simulated) groundwater data and a lot of nested catchments, where relationships between those catchments are clearly stated in the catchment attributes. The addition of many ungauged basins together with simulated discharge timeseries and catchment attributes gives the opportunity to study ungauged basins on a large scale.

Overall, I think the manuscript is a great addition to the growing pool of harmonized hydrological datasets. I hence recommend that the manuscript can be published after "minor" revisions: The only major comment I have is regarding the title:

- title: CAMELS datasets usually state the number of gauged stations in their title, with the number of 3330 Catchments in the title, I would assume that there is data for all of this Catchments available
    - also L103: "CAMELS-DK consistent data for over 3300 catchments" -> the data is not really consistent for all 3300 catchments, as some catchments have observed discharge while the majority does not provide observed discharge, a major difference across the catchments in the dataset

    Although the authors make it clear in their manuscript that the dataset includes ungauged and gauged catchments, I still think the title, particularly in comparison to other CAMELS datasets, is misleading.

**Specific (minor) comments**

- L19-L20: streamflow observations are typically the core component of CAMELS datasets, so in my opinion, these 304 catchments with streamflow data are really the core of CAMELS-DK, the enhancement maybe are the other 3026 catchments without observed streamflow
- L10-L25 (Abstract): maybe add the temporal range of your timeseries data to the abstract (1989-2019), as this is an essential information about the dataset?
- L126: Is it possible to cite the source of ID15v2.5 and state who / which institution made this division of Denmark into 3351 catchments?
- L158-L161: About the correction based on Stisen et al., 2011 -> I get that you conducted the correction of precipitation with several DMI data products yourself,

correct? Can you elaborate why sensors/gauges under-catch precipitation and why this correction is necessary? Also, can you briefly describe how the correction approach by Stisen et al. works?

- L164-L165: "DMI also provides gridded datasets of wind speed, air temperature and potential evapotranspiration (Scharling, 1999a). All variables are provided at daily timesteps; however, at an original resolution of 20 km2." -> By all variables you refer to wind speed, air temperature and potential evapotranspiration, correct? In Table 1 you only list temperature and pet, not wind speed. Also, did you calculate the pet with the Makkink formula yourself or has this already been done by the DMI? Perhaps you could expand on the section about the climate variables temperature, pet and wind speed and also indicate how you processed the data from the DMI (as you did in L154: "aggregated to catchment scale")

- L165: Give citation of the modified Makkink formula

- L151-165: Can you add a sentence about uncertainty in the climate data? Does the DMI state something about their uncertainty? Does the varying number of 200-500 stations over time affect uncertainty and does Stisen et al. correction affect uncertainty?

- L166 (Table 1):
  - Sources: better add the sources/citations of the datasets to the references and refer to them in the table, this way you can also add when you last visited the websites
  - Source for observed streamflow (Qobs) is the DK-model, should be Danish EPA / Aarhus University?
  - What is "SZ" in the description of some variables?
  - column Qsim is called Qdkm in the dataset

- L170: "Danish EPA" -> what is EPA?

- L171-L172: "Water levels are measured sub-daily (minutes)" -> how many minutes?

- L178: "station may have very limited time series length" -> what was the minimum time series length for CAMELS-DK?

- L186-L187: "Most of the stations have observed streamflow available during the entire years during 1989 to 2005" -> I do not fully comprehend this sentence, the entire years would be 1989-2019, or does this mean that these stations do not have any missing data between 1989 and 2005? You could also give a percentage number for "most of the stations"

- L185-L189: I think you could improve the section about data availability over time by expanding it further and structuring it more clearly. It doesn't really seem finished, and the incomplete sentence gives the impression that you actually wanted to write more here.

- L191 (Figure 2):
  - (a):
    - marker sizes in the legend do not match marker sizes on the map, the marker sizes on the map are good in my opinion, so just increase the marker sizes in the legend, especially for smallest markers.
    - Also, you could change the legend title "Area" to "Catchment area" to clarify what is meant here.
    - You could also think about switching to other colors, as the combination of red and green is not colorblind-friendly.
  - (b):

- caption: "number of stations have available streamflow" -> "number of stations that have available streamflow"
- to me, "have data over the entire year" suggests that there is no data missing for these stations over the entire year (all days have data), which should not be the case, looking at panel (c)
  - (c):
    - caption: "different period" -> "different periods"
    - you could also use different colors that are colorbild-friendly
- L210-L211: Why are the streamflow observations used for the DK-model only mostly similar to the timeseries in CAMELS-DK? Can you very briefly state the reasons for that here?
- L255: maybe cite the official source for the ID15 catchments, if possible?
- L257: "Flow direction indicates the downstream catchments, the column was filled with –9999 for most downstream catchments." -> the actual number in the dataset is -99
- L266 (Table 2):
  - unit of longitude and latitude is ° (degree)
  - column names for elevation are different in the dataset (dem_mean, dem_max, dem_median, dem_min instead of ele_mean, ele_max, ele_median, ele_min)
  - cite the 10m DEM via references in the Data source column if possible
- L271: Can you refer to formulas / code you used for the calculation of climate indices? You could also refer to the Github repository of Nans Addor, where you find the code he used to calculate the climate indices (**https://github.com/naddor/camels**)
- L272-L273: Did you also use a threshold of e.g. 5 % missing streamflow data in hydrological years for the inclusion of the year for the calculation of climate indices as in other CAMELS datasets?
- L272-L273: "climatic indices are calculated for water years from 1989 to 2019" -> there are different definitions of "water years", e.g. Oct-Sept, Sept-Aug, please clarify which period you used for a hydrological year in CAMELS-DK
- L275 (Table 3): frac_snow is actually called frac_snow_daily in the dataset
- L282-L283: do you also have a reference for the second nine signatures you mention here?
- L284: You should refer here to Appendix A, where the additional hydrological signatures are listed
- L284-L285: same as for climate indices; did you only use "complete hydrological years" (max. 5% missing streamflow data in a year) for the calculation of indices like other CAMELS datasets?
- L286: Units here are often mm/timestep, I think the unit should better be mm d$^{-1}$ here
- L290-L302: Maybe state something about the limitations and uncertainty in comparing CLC data from different years due to different methodology and satellite data, is it also like that for Basemap04?
- L320: Maybe change the section name to "Hydrogeology and geology"?
- L336: Maybe change the Table name to "Catchment attributes of hydrogeologic and geologic features"?
- L358-L370: I think it would be great if you would include the NSEs and KGEs of your simulation results for each catchment in the dataset, this way others could compare

their model result to yours, which improves the usage of CAMELS-DK as a benchmark dataset. This way, it would also be possible to see how trustworthy the modeled results for individual catchments are (just an idea for improvement)

- L390 (Figure 6):
  - You could change the colors red and green here to make to figure colorblind-friendly
  - caption: "Climatology of precipitation observed streamflow, and..." -> "Climatology of precipitation, observed streamflow, and..."?
  - I would also prefer a more meaningful y-axis label here ("dtp" -> e.g. "simulated shallow groundwater level")
- L393: "Fig. 6 shows the climatology (1990 - 2019) of precipitation, observed streamflow and depth to phreatic layers" -> so are these the mean values of precipitation, observed streamflow and depth to phreatic layers over time? Maybe clarify this in the text.
- L425 (Figure 7):
  - again, I would prefer more meaningful variable names for axis and scale labels (e.g. "groundwater abstraction" instead of "DKM_gwe")
  - the x-axis label for the years is missing
  - Is there a reason for the inverted arrangement of panel (b)? I think the figure would be tidier if the map was also on the left and the time series on the right.
  - The resolution / dpi of Figure 7 is noticeably worse than that of the other figures
- L437-L438: "The Python script of processing the time series and landscape attributes based on original datasets are provided in a folder named 'Python'." -> Is it possible that this Python script is incomplete? It is just a very small script for getting accumulated basins, and accessing the attribute files and the timeseries files. It is not the code for processing the original datasets for CAMELS-DK, by your wording I would expect e.g. code for processing the DMI meteorological source data to the CAMELS-DK timeseries
- L450: Where do you provide the original raster data? -> maybe rephrase, that you give instructions on how to access the source data in the Data availability section. As mentioned above, I also could not find Python scripts for data processing

**Dataset**

- In general you have lots of decimal places for all your variables, you could think about rounding to 2-3 decimal places
- the shapefiles "CAMELS_DK_304_gauging_catchment_boundaries.shp" and "CAMELS_DK_3330_catchment_boundaries.shp" yield different catchment boundaries for the same catch_id. I get that the 3330 shapefile contains the ID15 boundaries and the 304 shapefiles contains the ID15 boundary of the catchment + all upstream catchments, but you should clearly state that in the data description, as this can get confusing if someone is not aware of that or uses the wrong shapefile.
- the files CAMELS_DK_signature_obs_based.csv and CAMELS_DK_signature_sim_based.csv are actually the same

- CAMELS_DK_soil.csv uses column name "Id15_model" instead of "catch_id" for the catchment id, you should change that in the soil attributes file to be consistent across the dataset and make it easier to work with it
- the file CAMELS_DK_topography states via column "gauged_type" that there should be observed timeseries data for catch_ids [13210020, 13231401, 16200602, 32211251, 32230800, 35320416, 35321469, 57220534] but there are no csv files in Dynamics/Gauged_catchments/ for these IDs, instead I find observed timeseries data for catch_ids [13210113, 13231400, 16200624, 32230635, 35320540, 57220030, 71270810, 72300455], where the topographic attributes state it should not exist.
The former list of "gauged" IDs from the topographic attributes are also missing in the stations shapefile and the gauging catchments shapefile, so I guess your topographic attributes are incorrect / out of date?
- In the shapefile CAMELS_DK_georegion.shp you have a georegion with the name "Thy" which are just big rectangles covering Denmark and the surrounding area. I think you can delete that from the shapefile?
- topographic attributes column "gauge_record_pct": with a value of 100, I would expect no NaN values in the entire Qobs timeseries, but in your case you only refer to the time period for which you have discharge data (1989-2019) while the meteorological variables extend to 2023, so you always have NaNs at the end of the observed discharge timeseries. I think you should make that clear in the data description: "the time percentage with available observed streamflow" -> e.g. "the time percentage with available observed streamflow in the period 1989-01-02 - 2019-12-31"
- Data description:
  - L35-L36: "The hydrometeorological time series and landscape attributes for each catchment are calculated using the python package rioxarray." -> I think this is correct for the meteorological timeseries and landscape attributes, not the observed hydrological timeseries?
  - L102: Source for observed streamflow (Qobs) is the DK-model, from the manuscript the source should be Danish EPA / Aarhus University
  - L102 (Appendix A): column Qsim is called Qdkm in the dataset
  - L107 (Appendix B1): column names for elevation are different in the dataset (dem_mean, dem_max, dem_median, dem_min instead of ele_mean, ele_max, ele_median, ele_min)
  - L109 (Appendix B2): frac_snow is actually called frac_snow_daily in the dataset
  - L113 (Appendix B4): same comments as for technical manuscript comments L303
  - The tables in the appendix should contain all columns in the csv files, Appendix A misses the time column, Appendix B2-B6 miss the catch_id column

**Technical corrections**

- L12-L13: the acronym CAMELS actually stands for "Catchment Attributes and Meteorology for Large-sample Studies"

- L30: "basis of supporting water resource management..." -> "basis of support for the management of water resources..." or "basis for supporting water resources management..."
- L37: "Bing able" -> "Being able"
- L52: "catchments datasets" -> I think the word "datasets" can be removed here
- L55: "conined" -> "coined"
- L77-L78: "Especially, the lack samples from low-lying, small-size, and groundwater-dominated catchments." -> e.g. "Especially a lack of samples from low-lying, small and groundwater-dominated catchments."
- L83: "CAMLES" -> "CAMELS"
- L89: "CMAELS" -> "CAMELS"
- L90: "have already contribute" -> "have already contributed"
- L102-L103: webpages should better be added to the references, not cited in-text, in any case, add "last visited …" information
- L103: "CAMELS-DK consistent data for over 3300 catchments" -> e.g. "CAMELS-DK provides consistent data for over 3300 catchments"
- L107: "catchments attributes" -> "catchment attributes"
- L114: "CAMELS dataset" -> "CAMELS datasets"
- L119: "enrich existing CAMELS database" -> "enrich the existing CAMELS database"
- L120: "which respect to" -> "with respect to"
- L166 (Table 1):
  - Units: according to the ESSD submission guidelines, units must be written exponentially (e.g. "mm/d" -> "mm d$^{-1}$")
  - also the case for Table 2, 3, 4, 6, Appendix A, Figure 3, 6, 7
- L171: "through the surface water database of the (https://odaforalle.au.dk/)" -> sentence is not complete, also better to move urls / webpages to references or add "last visited …" to the url
- L175: "are accessible through (https://odaforalle.au.dk/)" -> again, better to move the url to the references and cite, e.g. "are accessible through the ODA platform (Aarhus University, 2024)", add "last visited …" in any case
- L187: "However, many hydrological stations ." -> incomplete sentence
- L215: again, better cite webpages via the references, add "last visited …"
- L219: "attention has been paid the representation" -> "attention has been paid to the representation"
- L250: "are also provide" -> "are also provided"
- L255: "The dataset is organized based on the ID15 catchment." -> "The dataset is organized based on the ID15 catchments."
- L255-L256: "eight-digital name" -> "eight-digit name"
- L262: "monitoring data of the data"? You mean percentage of available observed streamflow data in the time period?
- L273: "which is consistency to the availability of observed streamflow" -> "which is consistent with the availability of observed discharges."
- L303 (Table 5):
  - "Yeas" -> typo in the header
  - Description:
    - You could add "area" at the end of each description; e.g. "Percentage of agriculture" -> "Percentage of agricultural areas", "Percentage of urban" -> "Percentage of urban areas"

- - "Percentage of deciduous forest in" -> the "in" at the end is wrong
    - You have the full citation of Adhikari et al. in the Data sources and References column, move that to the References section at the end and just cite here
    - "Data source and References" column could be renamed to just "Data source" to be consistent with all other tables
- L304: "based on a regression modelling" -> "based on a regression model"
- L314-L316: "Several hydraulic parameters from the dataset of 3D Soil Hydraulic Database of Europe at 1 km and 250 m resolution, such as the water content at field capacity and saturated hydraulic conductivity (Table 6)." -> this sentence is not complete
- L336 (Table 7): First data source are two DOIs; add the full citation with DOIs to the references and cite here
- L353: (Figure 3):
    - order of labels is incorrect (a, b, c, d, f, e)
    - labels (b) and (f) overlap with plotted outliers -> move labels a little bit to the left or to the upper left corner outside the axes
- L401: "clay percentage in Fig. 6d" -> I think you want to refer to Fig. 3d here
- L415: "blow" -> "below"
- L420: again, better cite Jupiter via References
- L446: "The dataset is developed to assistant machine learning studies" -> "The dataset is developed to assist machine learning studies"
- L448: "ang" -> "and"
- L462: use exponential notation for units
- L463-L464: "indicating that a slightly smoother streamflow hydrograph simulated by the DK-model" -> "indicating a slightly smoother streamflow hydrograph simulated by the DK-model"
- L465: "challenged to capture accurately by the model" -> "challenged to be captured accurately by the model"
- L474-L475: "The ID15 catchment shapefile is provided with this dataset, previous version, ..." -> "The previous version X of the ID15 catchment shapefile is provided with this dataset, ..."

---

## Author Response (AR1)

**Dear Editor,**

We are pleased to submit the revised version of our manuscript titled *CAMELS-DK: Hydrometeorological Time Series* and Landscape Attributes for 3330 Danish Catchments with Streamflow Observations from 304 Gauged Stations, along with the updated dataset and data description. We have thoroughly addressed the questions and comments from all four reviewers. Their insightful feedback has greatly improved the quality and clarity of our work.

For clarity and ease of review, we have used black font for the reviewers' comments and blue font for our replies in the response document. We have also highlighted the corresponding changes in the revised manuscript where applicable. We thank you and the reviewers for your time and effort in evaluating our manuscript, and we look forward to your feedback on the revised version.

Best regards,

Jun Liu (on behalf of all co-authors)

**Response to Reviewer 1 (Ather Abbas)**

Thanks for the paper and data.

There is a similar dataset at https://zenodo.org/records/7962379 by one of the co-authors (Julian Koch). Can the authors please clarify that if this dataset is an updated version of the one provided at zenodo in terms of observed streamflow? Regards,

**Ather Abbas.**

**Reply**: Yes, the dataset you mention is related. The Caravan extension published on Zenodo is connected to a manuscript (https://doi.org/10.34194/geusb.v49.8292) and a dataset (https://doi.org/10.22008/FK2/YCQXTR). The current dataset associated to this manuscript, however, is more extensive, including both, simulated and observed data and the temporal coverage is updated and extends beyond 2019. Another major difference is that the current dataset contains information for all ~3300 catchments in Denmark, and not only the ~300 gauged catchments. In addition, the number of static catchment attributes has been expanded substantially.

**Response to Reviewer 2 (Alexander Dolich)**

**General comments**

Thank you for your work compiling the CAMELS-DK dataset and the manuscript. The introduction gives an excellent overview of the current state of large-sample hydrological datasets and the challenges of preprocessing and retrieving large amounts of harmonized hydro-meteorological data, which CAMELS datasets help to solve. The data discussion section gives a very good and helpful overview on the hydrologic characteristics and specificities of Denmark, which is especially helpful for someone who is not familiar with the hydrological landscape in Denmark and can provide great insights and ideas for the discussion of one's own study results.

CAMELS-DK is another great addition to the ever-growing family of CAMELS datasets, especially with the incorporation of (simulated) groundwater data and a lot of nested catchments, where relationships between those catchments are clearly stated in the catchment attributes. The addition of many ungauged basins together with simulated discharge timeseries and catchment attributes gives the opportunity to study ungauged basins on a large scale.

Overall, I think the manuscript is a great addition to the growing pool of harmonized hydrological datasets. I hence recommend that the manuscript can be published after "minor" revisions: The only major comment I have is regarding the title:

• title: CAMELS datasets usually state the number of gauged stations in their title, with the number of 3330 Catchments in the title, I would assume that there is data for all of this Catchments available

also L103: "CAMELS-DK consistent data for over 3300 catchments"-> the data is not really consistent for all 3300 catchments, as some catchments have observed discharge while the majority does not provide observed discharge, a major difference across the catchments in the dataset.

Although the authors make it clear in their manuscript that the dataset includes ungauged and gauged catchments, I still think the title, particularly in comparison to other CAMELS datasets, is misleading.

**Reply:** We thank the reviewer for his time and thoughtful feedback on our manuscript. The comments and suggestions provided have been invaluable in improving the quality and clarity of our work. In response to the reviewer's comment about the title, we acknowledge that CAMELS datasets typically include the number of gauged stations in their titles. To address this and ensure consistency with other CAMELS datasets, we have revised the title of our manuscript to:

"CAMELS-DK: Hydrometeorological Time Series and Landscape Attributes for 3330 Danish Catchments with Observed Streamflow from 304 Gauged Stations."

**Specific (minor) comments**

• L19-L20: streamflow observations are typically the core component of CAMELS datasets, so in my opinion, these 304 catchments with streamflow data are really the core of CAMELS-DK, the enhancement maybe are the other 3026 catchments without observed streamflow.

**Reply**: Yes, we agree that the core of CAMELS-DK is the observed streamflow from the 304 gauging stations. Accordingly, we have updated the title to emphasize the number of gauging stations.

• L10-L25 (Abstract): maybe add the temporal range of your timeseries data to the abstract (1989-2019), as this is an essential information about the dataset? **Reply**: We have included the time period in the abstract.

• L126: Is it possible to cite the source of ID15v2.5 and state who / which institution made this division of Denmark into 3351 catchments?

**Reply**: yes, we have added the reference to the ID15 shapefile.

• L158-L161: About the correction based on Stisen et al., 2011-> I get that you conducted the correction of precipitation with several DMI data products yourself, correct? Can you elaborate why sensors/gauges under-catch precipitation and why this correction is necessary? Also, can you briefly describe how the correction approach by Stisen et al. works? **Reply**: yes, we conducted the correction of precipitation. We added the reason of the correction and describe the approach by Stisen in the updated manuscript (line 163-171).

• L164-L165: "DMI also provides gridded datasets of wind speed, air temperature and potential evapotranspiration (Scharling, 1999a). All variables are provided at daily timesteps; however, at an original resolution of 20 km2."-> By all variables you refer to wind speed, air temperature and potential evapotranspiration, correct? In Table 1 you only list temperature and pet, not wind speed. Also, did you calculate the pet with the Makkink formula yourself or has this already been done by the DMI? Perhaps you could expand on the section about the climate variables temperature, pet and wind speed and also indicate how you processed the data from the DMI (as you did in L154: "aggregated to catchment scale") **Reply**: It was a mistake to mention wind speed in this context, as we have excluded this variable from our analysis. By "all variables," we were referring only to air temperature and potential evapotranspiration (PET), which is why wind speed is not listed in Table 1. Additionally, PET was calculated by the DMI using the Makkink formula, not by us. To address your comment, we have expanded the section on climate variables to provide more detail on how we processed the data from the DMI. Specifically, we have added the following lines to describe our methods (line 180-184)

The climate data was downloaded directly from the DMI Frie Data Application Programming Interface. Precipitation correction was then applied to all grids to account for the undercatch biases. The raster data was subsequently clipped to each catchment boundary, and a mean daily value was calculated based on all grids that fall within or touch the boundaries. This process forms the climate time series included in CAMELS-DK.

• L165: Give citation of the modified Makkink formula

**Reply**: the citation has been added.

• L151-165: Can you add a sentence about uncertainty in the climate data? Does the DMI state something about their uncertainty? Does the varying number of 200-500 stations over time affect uncertainty and does Stisen et al. correction affect uncertainty?

**Reply**: To our knowledge, DMI does not provide detailed information about the general uncertainty in the climate data. However, the impact of the reduction in the number of precipitation gauges over time was evaluated by DMI [reference added to the manuscript] and found to be minimal at the national scale but with potential for localized impacts. We have added the following information to the updated manuscript: "The impact of the reduction in the number of gauges was evaluated by DMI and found to be minimal at the national scale but with potential for localized impacts (Andersen, 2021)."

Regarding the precipitation correction, it does introduce some additional uncertainty, particularly after 2010, when the sensor network became heterogeneous, incorporating three different gauge types. Ideally, the correction factors should be gauge type-dependent, but this information is unavailable, which may add to the uncertainty in the corrected data.

**• L166(Table 1):**

• Sources: better add the sources/citations of the datasets to the references and refer to them in the table, this way you can also add when you last visited the websites

**Reply**: we have added the sources/citations of the datasets to the references.

• Source for observed streamflow (Qobs) is the DK-model, should be Danish EPA / Aarhus University?

**Reply**: we have changed the confirmed the source.

• What is "SZ" in the description of some variables?

Reply: It is 'saturated zone', which has been corrected

o column Qsim is called Qdkm in the dataset

Reply: 'Qdkm' is used, table 1 and Appendix A in data description has been changed.

• L170: "Danish EPA"-> what is EPA?

Reply: EPA stands for the Danish Environmental Protection Agency, which has been added to the manuscript.

**• L171-L172: "Water levels are measured sub-daily (minutes)"-> how many minutes?**

**Reply**: The exact measurement interval in minutes varies depending on the sensors, and this information is not consistently available. To avoid potential misunderstanding, we have removed the reference to "minutes" in the text.

• L178: "station may have very limited time series length"-> what was the minimum time series length for CAMELS-DK?

**Reply**: The minimum time series length in CAMELS-DK is 1,762 days (corresponding to a gauge\_record\_pct of 15.56%). This information has been added to the manuscript (line 194-196) as follows:

"However, many stations are unsuited for use in hydrological modelling. Some stations may have limited time series lengths (basin\_id 35321223, which has the shortest streamflow record of 1,762 days during the period 1989–2019 in CAMELS-DK), some stations may contain questionable discharge values."

• L186-L187: "Most of the stations have observed streamflow available during the entire years during 1989 to 2005"-> I do not fully comprehend this sentence, the entire years would be 1989-2019, or does this mean that these stations do not have any missing data between 1989 and 2005? You could also give a percentage number for "most of the stations" **Reply**: We have rewritten the section to provide a clearer and more structured explanation of data availability. Please see the revised text below or in the updated manuscript.

• L185-L189: I think you could improve the section about data availability over time by expanding it further and structuring it more clearly. It doesn't really seem finished, and the incomplete sentence gives the impression that you actually wanted to write more here.

Reply: we have rewritten the section in the updated manuscript, see lines 204 -214.

• L191(Figure 2):

• (a):

■ marker sizes in the legend do not match marker sizes on the map, the marker sizes on the map are good in my opinion, so just increase the marker sizes in the legend, especially for smallest markers.

Reply: We have adjusted the marker sizes on the map.

Also, you could change the legend title "Area" to "Catchment area" to clarify what is meant here.

Reply: We have changed the legend title as "Catchment area".

■ You could also think about switching to other colors, as the combination of red and green is not colorblind-friendly.

Reply: we changed the colours to colourblind-friendly colours.

• (b):

■ caption: "number of stations have available streamflow"-> "number of stations that have available streamflow" **Reply**: we have changed the caption.

• to me, "have data over the entire year" suggests that there is no data missing for these stations over the entire year (all days have data), which should not be the case, looking at panel (c)

**Reply**: Panel (b) shows availability of streamflow observations per single year; panel (c) shows it for different ranges of years. We changed the figure caption to make this clearer.

○ (c):

■ caption: "different period"-> "different periods"

**Reply**: we have changed the caption.

■ you could also use different colors that are colorbild-friendly

Reply: we changed the colours to colourblind-friendly colours.

• L210-L211: Why are the streamflow observations used for the DK-model only mostly similar to the timeseries in CAMELS-DK? Can you very briefly state the reasons for that here?

**Reply**: The number of stations used in the DK-model calibration changes slightly over time, dependent on changing data availability through the years, changing calibration periods etc. Here, we present data from 304 stations, which is an additional 3 stations compared to the latest DK-model calibration.

• L255: maybe cite the official source for the ID15 catchments, if possible?

**Reply**: we have added the reference to the updated manuscript.

• L257: "Flow direction indicates the downstream catchments; the column was filled with –9999 for most downstream catchments."-> the actual number in the dataset is -99

Reply: you are right, '-99' represents the catchments are downstream catchments.

• L266(Table 2):

 $\circ$  unit of longitude and latitude is  $^{\circ}$  (degree)

o column names for elevation are different in the dataset (dem\_mean, dem\_max, dem\_median, dem\_min instead of ele\_mean, ele\_max, ele\_median, ele\_min)

 $\circ$  cite the 10m DEM via references in the Data source column if possible

**Reply**: We have added the units for longitude and latitude. Since we used UTM zone 32N as the coordinate system, the unit is meters. Additionally, we have updated the variable names to elev\_mean, elev\_max, elev\_median, and elev\_min to ensure consistency with other CAMELS datasets.

• L271: Can you refer to formulas / code you used for the calculation of climate indices? You could also refer to the Github repository of Nans Addor, where you find the code he used to (https://github.com/naddor/camels) calculate the climate indices

**Reply:** The climate indices were calculated using the script developed by Hao et al. (2021). This information has been added to the updated manuscript.

• L272-L273: Did you also use a threshold of e.g. 5 % missing streamflow data in hydrological years for the inclusion of the year for the calculation of climate indices as in other CAMELS datasets? **Reply**: no, we did not set the threshold.

• L272-L273: "climatic indices are calculated for water years from 1989 to 2019"-> there are different definitions of "water years", e.g. Oct-Sept, Sept-Aug, please clarify which period you used for a hydrological year in CAMELS-DK **Reply**: It is incorrect the use 'water year'. The climate indices are calculated at a daily time scale.

• L275(Table 3): frac\_snow is actually called frac\_snow\_daily in the dataset **Reply**: we have changed the name in the dataset.

• L282-L283: do you also have a reference for the second nine signatures you mention here? **Reply**: References and the source code for the hydrological signatures have been added to the manuscript. Additionally, the description of the hydrological and additional signatures has been clarified and rephrased in Section 4.3.

• L284: You should refer here to Appendix A, where the additional hydrological signatures are listed **Reply**: Appendix A has been referred in the updated manuscript.

• L284-L285: same as for climate indices; did you only use "complete hydrological years" (max. 5% missing streamflow data in a year) for the calculation of indices like other CAMELS datasets? **Reply**: no, we did not set the threshold.

• L286: Units here are often mm/timestep, I think the unit should better be mm d-1 here **Reply**: Yes, we have corrected the typo.

• L290-L302: Maybe state something about the limitations and uncertainty in comparing CLC data from different years due to different methodology and satellite data, is it also like that for Basemap04?

**Reply**: we agree that there are limitations and uncertainties associated with CLC data due to differences in methodology and satellite data across years, we think that discussion of these aspects falls outside the scope of this paper. Since we present the CLC data as percentages of area, the impact of these uncertainties is minimized and does not significantly affect the overall interpretation or usage of the dataset. For this reason, we have not included a statement on these limitations in the manuscript.

**• L320: Maybe change the section name to "Hydrogeology and geology"?**

Reply: We have changed the section name.

• L336: Maybe change the Table name to "Catchment attributes of hydrogeologic and geologic features"?

**Reply: We have changed the name of the Table.**

• L358-L370: I think it would be great if you would include the NSEs and KGEs of your simulation results for each catchment in the dataset, this way others could compare their model result to yours, which improves the usage of CAMELS-DK as a benchmark dataset. This way, it would also be possible to see how trustworthy the modelled results for individual catchments are (just an idea for improvement).

**Reply**: We decided not to include NSE and KGE values in the dataset itself. We already provide the time series of both observed discharge (Qobs) and simulated discharge (Qdkm) in CSV files as separate columns, making it straightforward for users to calculate NSE, KGE, or other performance metrics for their specific periods of interest. Additionally, Fig. 4 in the manuscript already illustrates the spatial distribution of NSE and KGE values, providing an overview of model performance across the catchments. This ensures flexibility while maintaining the utility of CAMELS-DK as a benchmark dataset.

**• L390(Figure 6):**

• You could change the colours red and green here to make to figure colorblind-friendly

o caption: "Climatology of precipitation observed streamflow, and..."-> "Climatology of precipitation, observed streamflow, and..."?

• I would also prefer a more meaningful y-axis label here ("dtp"-> e.g. "simulated shallow groundwater level") **Reply**: we have changed the figure accordingly, including using colourblind-friendly colours, adding comma to the caption, and using 'shallow groundwater level (m)' instead of abbreviation.

• L393: "Fig. 6 shows the climatology (1990-2019) of precipitation, observed streamflow and depth to phreatic layers"-> so are these the mean values of precipitation, observed streamflow and depth to phreatic layers over time? Maybe clarify this in the text.

Reply: We have added the following sentence of how the climatology was calculated to the updated manuscript:

'Climatology is calculated by averaging time series values for the same day of the year across 1990-2019.'

• L425(Figure 7):

• again, I would prefer more meaningful variable names for axis and scale labels (e.g. "groundwater abstraction" instead of "DKM gwe")

Reply: We have modified Figure 7 accordingly; the axis names and scale labels have been updated.

the x-axis label for the years is missing**Reply**: We have added 'Time' as the x-axis label.

• Is there a reason for the inverted arrangement of panel (b)? I think the figure would be tidier if the map was also on the left and the time series on the right.

**Reply**: There was no specific reason for arranging panel (b) differently. We agree that the maps should be on one side and the time series on the other. In the updated manuscript, we have placed the time series from the same catchment on the left side and the catchment-aggregated values on the right side.

**• The resolution / dpi of Figure 7 is noticeably worse than that of the other figures**

**Reply**: Figure 7 is large in figure size, and it was compressed when inserted into Word. We have adjusted the settings to ensure high fidelity in Word, and we will provide the figure directly in the future.

• L437-L438: "The Python script of processing the time series and landscape attributes based on original datasets are provided in a folder named 'Python'."-> Is it possible that this Python script is incomplete? It is just a very small script for getting accumulated basins, and accessing the attribute files and the timeseries files. It is not the code for processing the original datasets for CAMELS-DK, by your wording I would expect e.g. code for processing the DMI meteorological source data to the CAMELS-DK timeseries.

Reply: We have added more functions in 'Python' script, including the raster clipping and processing attributes data.

• L450: Where do you provide the original raster data?-> maybe rephrase, that you give instructions on how to access the source data in the Data availability section. As mentioned above, I also could not find Python scripts for data processing.

**Reply**: We have rephrased the sentence to clarify that while we are not permitted to share most of the source data in their original formats, we provide detailed instructions in the Data Availability section on how to access these datasets. Additionally, we have included Python functions to guide users in processing the source data.

**Dataset**

• In general you have lots of decimal places for all your variables, you could think about rounding to 2-3 decimal places **Reply**: We have updated the dataset by rounding all attribute values to 3 decimal places. • the shapefiles "CAMELS\_DK\_304\_gauging\_catchment\_boundaries.shp" and "CAMELS\_DK\_3330\_catchment\_boundaries.shp" yield different catchment boundaries for the same catch\_id. I get that the 3330 shapefile contains the ID15 boundaries and the 304 shapefiles contains the ID15 boundary of the catchment + all upstream catchments, but you should clearly state that in the data description, as this can get confusing if someone is not aware of that or uses the wrong shapefile.

**Reply**: we have emphasis the differences of the 304 shapefile and the 3330 shapefile in the data description as follows: *Please note that the boundaries of the 304 gauged catchments are delineated using the ID15 catchments shapefile, with the downstream catchment identified by its catch\_id. The script for generating catchment boundaries is provided in the Python folder.*

• the files CAMELS\_DK\_signature\_obs\_based.csv, CAMELS\_DK\_signature\_sim\_based.csv are actually the same and **Reply:** we have corrected the mistake.

• CAMELS\_DK\_soil.csv uses column name "Id15\_model" instead of "catch\_id" for the catchment id, you should change that in the soil attributes file to be consistent across the dataset and make it easier to work with it **Reply**: the typo has been corrected.

• the file CAMELS\_DK\_topography states via column "gauged\_type" that there should be observed timeseries data for catch\_ids [13210020, 13231401, 16200602, 32211251, 32230800, 35320416, 35321469, 57220534] but there are no csv files in Dynamics/Gauged\_catchments/ for these IDs, instead I find observed timeseries data for catch\_ids [13210113, 13231400, 16200624, 32230635, 35320540, 57220030, 71270810, 72300455], where the topographic attributes state it should not exist. The former list of "gauged" IDs from the topographic attributes are also missing in the stations shapefile and the gauging catchments shapefile, so I guess your topographic attributes are incorrect / out of date?

**Reply**: We have corrected the inconsistencies in the CAMELS\_DK\_topography table to ensure that the "gauged\_type" column accurately reflects the availability of observed timeseries data. The updated table now aligns with the stations shapefile, gauging catchments shapefile, and the observed timeseries data files in Dynamics/Gauged\_catchments/.

• In the shapefile CAMELS\_DK\_georegion.shp you have a georegion with the name "Thy" which are just big rectangles covering Denmark and the surrounding area. I think you can delete that from the shapefile?

**Reply**: The previous georegion covered a larger area, including the ocean. We have now masked the georegion using the land boundary of Denmark. The region 'Thy' represents a large area in northwest Denmark.

• topographic attributes column "gauge\_record\_pct": with a value of 100, I would expect no NaN values in the entire Qobs timeseries, but in your case you only refer to the time period for which you have discharge data (1989-2019) while the meteorological variables extend to 2023, so you always have NaNs at the end of the observed discharge timeseries. I think you should make that clear in the data description: "the time percentage with available observed streamflow"-> e.g. "the time percentage with available observed streamflow in the period 1989-01-02 2019-12-31"

**Reply**: We have clarified the time period in the data description by specifying it in Table 2 of the manuscript and in Appendix B1 in data description. This now explicitly states that the time percentage refers to the period 1989-01-02 to 2019-12-31.

**Data description**

• L35-L36: "The hydrometeorological time series and landscape attributes for each catchment are calculated using the python package rioxarray."-> I think this is correct for the meteorological timeseries and landscape attributes, not the observed hydrological timeseries?

**Reply**: here we changed the 'hydrometeorological' to 'meteorological', streamflow data was read by pandas from csv files. We have rewrite the sentence see lines 34-37.

L102: Source for observed streamflow (Qobs) is the DK-model, from the manuscript the source should be Danish EPA
 / Aarhus University

**Reply**: we have corrected the reference here.

o L102(Appendix A): column Qsim is called Qdkm in the dataset

Reply: we use 'Qdkm' here, and the manuscript table 1, which is the same as dataset.

• L107 (Appendix B1): column names for elevation are different in the dataset (dem\_mean, dem\_max, dem\_median, dem\_min instead of ele\_mean, ele\_max, ele\_median, ele\_min)

**Reply**: the column names for elevation has been connected (we use 'elev\_') in the manuscript, dataset description and dataset.

o L109 (Appendix B2): frac\_snow is actually called frac\_snow\_daily in the dataset

Reply: The dataset has been corrected.

o L113 (Appendix B4): same comments as for technical manuscript comments L303

**Reply**: we have correct the table accordingly.

• The tables in the appendix should contain all columns in the csv files, Appendix A misses the time column, Appendix B2-B6 miss the catch\_id column

**Reply**: We have added the missing time column to Appendix A and the catch\_id column to Appendices B2–B6 to ensure that the tables now include all columns from the corresponding CSV files.

**Technical corrections**

• L12-L13: the acronym CAMELS actually stands for "Catchment Attributes and Meteorology for Large-sample Studies"

Reply: It has been corrected.

• L30: "basis of supporting water resource management..."-> "basis of support for the management of water resources..." or "basis for supporting water resources management..."

Reply: the sentence has been corrected.

• L37:"Bing able"-> "Being able"

Reply: corrected.

• L52:"catchments datasets"-> I think the word "datasets" can be removed here **Reply**: the word 'datasets' has been removed.

• L55:"conined"-> "coined" **Reply**: corrected.

• L77-L78: "Especially, the lack samples from low-lying, small-size, and groundwater-dominated catchments."-> e.g. "Especially a lack of samples from low-lying, small and groundwater-dominated catchments." **Reply**: the sentence has been changed accordingly."

• L83:"CAMLES"-> "CAMELS" Reply: corrected

• L89:"CMAELS"-> "CAMELS" Reply: corrected

• L90:"have already contribute"-> "have already contributed" **Reply**: corrected

• L102-L103: webpages should better be added to the references, not cited in-text, in any case, add "last visited ..." information

Reply: the webpages have been added to the references.

• L103: "CAMELS-DK consistent data for over 3300 catchments"-> e.g. "CAMELS-DK provides consistent data for over 3300 catchments"

Reply: corrected.

• L107: "catchments attributes"-> "catchment attributes" **Reply**: corrected.

• L114: "CAMELS dataset"-> "CAMELS datasets" Reply: corrected

• L119: "enrich existing CAMELS database"-> "enrich the existing CAMELS database"

**Reply: corrected**

• L120: "which respect to"-> "with respect to"

**Reply: corrected**

• L166(Table 1):

• Units: according to the ESSD submission guidelines, units must be written exponentially (e.g. "mm/d"-> "mm d-1")
• also the case for Table 2, 3, 4, 6, Appendix A, Figure 3, 6, 7 **Reply**: we have modified the units in Table 1, 2, 3, 4, 6, Appendix A, Figure 3, 6, 7, and units mentioned in the tests.

• L171: "through the surface water database of the (https://odaforalle.au.dk/)"-> sentence is not complete, also better to move urls / webpages to references or add "last visited ..." to the url **Reply**: the URLs has been put to references.

• L175: "are accessible through (https://odaforalle.au.dk/)"-> again, better to move the url to the references and cite, e.g. "are accessible through the ODA platform (Aarhus University, 2024)", add "last visited ..." in any case **Reply**: reference added.

• L187: "However, many hydrological stations ."-> incomplete sentence **Reply**: The sentence has been completed.

• L215: again, better cite webpages via the references, add "last visited ..." Reply: We have moved the referenced webpages to the references section and included "last visited [date]" for each.

• L219: "attention has been paid the representation"-> "attention has been paid to the representation" **Reply**: correct.

• L250: "are also provide"-> "are also provided" **Reply**: correct.

• L255: "The dataset is organized based on the ID15 catchment."-> "The dataset is organized based on the ID15 catchments."

Reply: correct.

• L255-L256: "eight-digital name"-> "eight-digit name" **Reply**: correct.

• L262: "monitoring data of the data"? You mean percentage of available observed streamflow data in the time period? **Reply**: yes, we have changed the phrase as 'such as the length and the percentage of available observed streamflow data during 1989 to 2019'

• L273: "which is consistency to the availability of observed streamflow"-> "which is consistent with the availability of observed discharges."

Reply: corrected.

• L303(Table 5):

o "Yeas"-> typo in the header

**Reply: corrected.**

 $\circ$  Description:

■ You could add "area" at the end of each description, e.g. "Percentage of agriculture"-> "Percentage of agricultural areas", "Percentage of urban"-> "Percentage of urban areas" Reply: corrected.

"Percentage of deciduous forest in"-> the "in" at the end is wrong
 Reply: corrected.

• You have the full citation of Adhikari et al. in the Data sources and References column, move that to the References section at the end and just cite here **Reply**: the references in 'Data sources' column have been cleaned and only the citation left.

• "Data source and References" column could be renamed to just "Data source" to be consistent with all other tables **Reply**: corrected.

• L304: "based on a regression modelling"-> "based on a regression model" **Reply**: corrected.

• L314-L316: "Several hydraulic parameters from the dataset of 3D Soil Hydraulic Database of Europe at 1 km and 250 m resolution, such as the water content at field capacity and saturated hydraulic conductivity (Table 6)."-> this sentence is not complete.

**Reply**: The sentence has been completed as 'Several hydraulic parameters from the 3D Soil Hydraulic Database of Europe, available at 1 km and 250 m resolutions, are included in CAMELS-DK, such as water content at field capacity and saturated hydraulic conductivity (Table 6).' In the updated manuscript.

**• L336 (Table 7): First data source are two DOIs; add the full citation with DOIs to the references and cite here**

**Reply**: The full citations, including the DOIs, have been added to the references section and are now cited in Table 7 as suggested in the updated manuscript.

• L353: (Figure 3):

 $\circ$  order of labels is incorrect (a, b, c, d, f, e)

 $\circ$  labels (b) and (f) overlap with plotted outliers-> move labels a little bit to the left or to the upper left corner outside the axes

**Reply**: we have correct the order of labels, and the labels have been moved to upper centre to avid overlapping with outliers.

• L401: "clay percentage in Fig. 6d"-> I think you want to refer to Fig. 3d here **Reply**: the error has been corrected.

• L415: "blow"-> "below" Reply: The typo has been corrected.

• L420: again, better cite Jupiter via References **Reply**: references used in the updated manuscript.

L446: "The dataset is developed to assistant machine learning studies"-> "The dataset is developed to assist machine learning studies"
 Reply: The typo has been corrected.

• L448: "ang"-> "and" **Reply**: The typo has been corrected.

• L462: use exponential notation for units **Reply**: The sentence has been rewritten.

• L463-L464: "indicating that a slightly smoother streamflow hydrograph simulated by the DK-model"-> "indicating a slightly smoother streamflow hydrograph simulated by the DK-model" **Reply**: The sentence has been rewritten.

• L465: "challenged to capture accurately by the model"-> "challenged to be captured accurately by the model" **Reply**: The sentence has been rewritten.

• L474-L475: "The ID15 catchment shapefile is provided with this dataset, previous version, ..."-> "The previous version X of the ID15 catchment shapefile is provided with this dataset, ... Reply: The sentence has been rewritten.

**Response to Reviewer 3 (Vazken Andréassian)**

Dear colleagues,

I read your paper and downloaded the dataset. Congratulations for this huge work.

While reading the data, I have found a case where it seems that there has been a conflation between the missing value code and the zero value : catchment 42600042, calendar year 2010 and 2011 (which follow a period with missing values). Could you please check? I may have made a mistake while reading the data... if so please accept my apologies. Best regards,

Vazken Andréassian

**Reply**: Thank you for your interest in our dataset. We have reviewed the zero values for catchment 42600042 during the years 2010 and 2011. This period appears unusual, as the source data shows zero values throughout. We will reach out to the data provider, and we agreed that the 0 values are a mistake. We have removed the 0 values in the revised dataset, and the year 2010 and 2011 have no observations in catchment 42600042.

**Response to Reviewer 4 (Ashutosh Sharma)**

The manuscript describes the CAMELS-DK dataset, which consists of hydrometeorological time series and catchment attributes for a large sample of catchments in Denmark. I appreciate the valuable contribution of the authors in extending the CAMELS dataset family to Denmark and providing a valuable dataset for the hydrological community. However, the manuscript requires substantial revision before it can be published.

One of the major concerns is the lack of observed streamflow data for many of the catchments. As described in the manuscript, the observed streamflow data is available only for the 304 catchments out of 3330 mentioned in the Title. A key characteristic of CAMELS datasets is providing observed streamflow data – without this, the datasets are less useful for large-sample hydrological analyses. I appreciate that the authors have included a modeled flow time series. However, I recommend keeping 304 catchments in the Title, i.e., where the authors have calculated hydrological signatures based on the observed data.

**Reply:** We understand the importance of observed streamflow data in hydrological analyses and the centrality of this feature to the CAMELS framework. The CAMELS-DK dataset was designed to provide a comprehensive resource for hydrological research in Denmark, and while observed streamflow data is limited to 304 catchments, the dataset includes modelled streamflow time series for all 3330 catchments.

We agree with the reviewer's suggestion to align the title with the availability of observed streamflow data. However, our intention was to emphasize the full spatial extent of the dataset, while explicitly detailing the limitations regarding observed streamflow in the manuscript. To address this concern, we propose revising the title to clearly reflect the distinction between observed and modelled data, the modified title is: *CAMELS-DK: Hydrometeorological Time Series and Catchment Attributes for 3330 Catchments in Denmark, Including Observed Streamflow for 304 Catchments*

This title acknowledges the full dataset while specifying the subset of catchments with observed streamflow. Additionally, we will emphasize in the abstract and introduction that observed streamflow is only available for 304 catchments, and modelled data is provided for the remaining catchments.

We hope this clarification and title revision adequately address the reviewer's concerns. Thank you again for the constructive feedback, which has helped us improve the clarity and focus of our manuscript.

**Specific comments:**

1) Title: The authors should keep the number of gauged stations (i.e., 304) in the Title of the manuscript to be consistent with other CAMELS datasets.

**Reply:** We have changed the title to: *CAMELS-DK: Hydrometeorological Time Series and Catchment Attributes for 3330 Catchments in Denmark, Including Observed Streamflow for 304 Catchments.*

2) Abstract: Line 17-22 can be rephrased to clearly distinguish that CAMELS-DK provides dynamic and static variables, including observed streamflow and their temporal range for 304 catchments, and provides dynamic and static variables for an additional 3026 catchments for further applications. Also, can the author mention how the additional catchments can enhance applicability without observed streamflow data? I understand that modeled streamflow is provided, but I am not convinced about its applicability given many catchments are very small in area, limited model resolution, and missing validation.

**Reply:** We have modified the abstract to distinguish that CAMELS-DK provides dynamic and static variables, including observed streamflow and their temporal range for 304 gauged catchments. Regarding the applicability of the model simulations for ungauged catchments, we have clarified in the abstract that they can be utilized in various contexts, such as the development of data-driven and hybrid physics-informed modeling frameworks, or other applications requiring consistent full spatial coverage across a larger number of catchments. We acknowledge the limitations associated with smaller catchment areas, model resolution, and missing validation, and have framed these simulations as complementary rather than a direct substitute for observed streamflow data.

3) Line 44-48: Please include the access dates of the cited weblinks.

**Reply:** We have added the access dates for all the cited weblinks in the manuscript.

4) There are several typos throughout the manuscript. I am restricting myself from providing a complete list here.However, the authors should thoroughly proofread the manuscript to correct the typos and grammatical mistakes.Reply: We have thoroughly proofread the manuscript to identify and correct typos and grammatical errors.

5) Line 103: "consistent data for 3300 catchments"? – As described in the manuscript, all data is not consistent for all 3330 catchments. Maybe "consistent meteorological and static attribute data for 3330 catchments? Moreover, is it 3300 or 3330? Please be specific.

Reply: We have corrected the sentence because CAMELS-DK has 3330 catchments.

6) Line 126: Please include a reference for ID15v2.5. Also, if ID15v2.5 includes 3351 catchments, why do the authors include 3330 in the manuscript? Given that many of these catchments are very small, how do authors justify the reliability of the model performance with 500 m resolution data [Line 208]?

**Reply**: We have added the reference for ID15v2.5 to the manuscript. Regarding the discrepancy in the number of catchments, we have clarified that 21 basins, primarily small islands, were excluded from the analysis due to their limited hydrological relevance. As for the concern about the 500 m resolution data, the surface module of our model includes 2D overland flow, a simplified two-layer description of the unsaturated zone, and 1D kinematic routing of streamflow. This approach ensures that the resolution has less impact on discharge simulations. Further details about the model structure can be found in Section 3.3, and Figure 4 provides an overview of the model's performance across catchments.

7) There are several mistakes in the figures and visualization. For example, Figure 1(b): "Catchment area" instead of "Area"; Figure 2(a): the size of the circles is enlarged in the legend; Figure 2(c) is not colorblind-friendly; Figure 3(a): "ele mean" instead of "DEM".

**Reply**: We have carefully reviewed and revised the figures in the updated manuscript to address the concerns raised: Figure 1(b): The label "Area" has been updated to "Catchment area" for consistency and clarity.

Figure 2(a): The size of the circles in the legend has been corrected to align with the actual data points in the figure. Figure 2(a, c) and Figure 6,: We have updated the color scheme to a colorblind-friendly palette to improve accessibility. Figure 3(a): The label "DEM" has been replaced with "ele\_mean" to accurately reflect the variable name used in the dataset. 8) Line 359-360: "CAMELS-DK provides observed streamflow data from 304 hydrological stations and simulated streamflow data for 2,942 catchments" – which makes it a total of 3246 catchments?; and it mentions, "However, 388 catchments lack observed or simulated streamflow data." – then why are those relevant/included in the CAMELS-DK dataset?

**Reply**: Thank you for your comment. The small catchments without apparent surface flow discharge are retained in the dataset because they contribute to water exchange with adjacent catchments. These catchments can be useful for attributing hydrological characteristics to downstream catchments, which is valuable for comprehensive hydrological analysis and modelling. Therefore, we have included them in the CAMELS-DK dataset despite the lack of observed or simulated streamflow data.

9) Can authors include more discussion on the DK-Model performance, especially the causes of deteriorated performance in the central and northern regions? Since the authors provide a modelled streamflow based on this model, the readers/users must understand the limitations.

**Reply**: we have added the following paragraph to discuss the model performance in the updated manuscript (lines:397-407).

10) Line 437-438 and Line 450: The Python script for processing the time series, landscape attributes, and original raster data is not included in the dataset. I suggest including the Python script for data processing or adding more details on data processing, detailing the different products and steps used in generating the dataset to build user confidence. I would also recommend adding the license/disclaimer as a text file within the dataset to ensure this is readily available when a user directly downloads the product.

**Reply**: We have modified the Python script in the dataset that can be used to process climate data, model simulations, and landscape attributes, including soil, land use, and geology features. However, we do not have the rights to directly share the source data used in the processing. Users will need to download the source data themselves from their respective providers.

11) There are several mistakes in the naming convention in the dataset. For example, CAMELS\_DK\_soil.csv uses column name "Id15\_model" instead of "catch\_id"; "CAMELS\_DK\_signature\_obs\_based.csv" and "CAMELS\_DK\_signature\_sim\_based.csv" includes the same data. I suggest checking all the files and making necessary corrections thoroughly.

**Reply**: Thank you for pointing this out. We have thoroughly reviewed the attribute tables and made the necessary corrections. Specifically, we have: Changed the column name "Id15\_model" to "catch\_id" in the file CAMELS\_DK\_soil.csv. Ensured that hydrological signatures based on observations are correctly saved in the file CAMELS\_DK\_signature\_obs\_based.csv. We have also double-checked the dataset to ensure consistency and accuracy across all files.

---

## Author Response (AR2)

Dear Editor,

Thank you for your comments on the manuscript, which have helped us improve its quality. We are sorry that we could not resubmit the modified manuscript before Christmas break. We planned to keep the dataset available on GEUS repository. The problem of GEUS repository has now been resolved, and we have tested that the dataset can be successfully downloaded from the website via the link. Our responses to the other comments are highlighted in blue.

Best regards,

Jun Liu (on behalf of all co-authors)

I cannot access the data at the GEUS repository. For single files, I receive the following error:

{"status":"ERROR","code":404,"message":"Datafile 84632: Failed to locate and/or open physical file."}

For downloading the full dataset, I receive an empty zip file. Please carefully check that the repository is stable, complete, and up to date. I would also guess that linking your paper will be a good addition. If the title can be updated, it might be worth considering.

**Reply**: There were some issues with the repository, which have been fixed now. We have tested the dataset can be downloaded from the link:

I find your data availability statement confusing, plus GEUS is misspelled there. Here is a suggestion:

The CAMELS-DK dataset is available at the GEUS Dataverse under a Creative Commons Zero license at https://doi.org/10.22008/FK2/AZXSYP (Koch et al., 2024). Since all data is continuously updated by the respective providers, we list the respective sources of their free availability: ID15 catchment shapefile is updated. The newest version of the shapefile can be requested by writing to id15@ecos.au.dk. High-resolution DEM data is available here https://dataforsyningen.dk/data/928. The gridded climate data is created by the Danish Meteorological Institute (DMI) and can be downloaded through the DMI Data API (https://opendatadocs.dmi.govcloud.dk/DMIOpenData). The observed streamflow data can be downloaded from the Danish Environmental Protection Agency (https://mst.dk/erhverv).

Simulation results of surface water and groundwater dynamics from the National Hydrological model of Denmark are available at (https://dennationalehydrologiskemodel.dk/).

**Reply**: we have modified the section, see lines: 520 – 528.

There are missing spaces (e.g. L23, L297, L436, and many more) and at least one twisted abbreviation (L514). This gives the manuscript a sloppy appearance. Please carefully correct this. This will also ease the further copyediting with Copernicus.

**Reply**: we have carefully checked the space errors and typos of the manuscript.

L128ff: ID15v2.5 vs. ID15 I find this very confusing. Obviously, there are several versions of the basins. But would it suffice to clarify this once and stick to the term "ID15-catchments"? L280 would be a good spot to refer to the specific version again as part of the citation.

**Reply**: agreed, we have uniformed the name of ID15 catchments as 'ID15 catchments' in the manuscript, dataset, and dataset description.

L228ff: Please carefully check the argumentation structure of these paragraphs. I find it hard to follow. Moreover, I am under the impression that the numbers are merely to show off. Could the numbered list L253ff simply become such a list? As block text, I get lost. And I have difficulties keeping track of the reasoning for each of the 8 simulated variables as part of the CAMELS-DK dataset. Exactly this could guide the argumentation in this subsection?

**Reply**: We removed some of the details surrounding the calibration of the DK-model, following your suggestion that they might partly have been unnecessary information here. Our argumentation in the latter part is that, as streamflow is tightly coupled with groundwater in Denmark, it is relevant to include groundwater-related variables (and others, such as soil moisture) from the DK-model. We cleaned up that section a bit, removing some details, hoping to make this more clear to the reader, the corrections can be found in lines: 258-277

I am even more confused by the setting that the model is used at 500 m lateral resolution but in L356 you refer to the 100 m grid plus vertical definitions comprised in 5 variables. Why did you separate subsection 3.3 from 4.6? I know that my comments are very late (and I am sorry for this), but I hoped that the comments of the referees would solve this.

**Reply**: We understand your confusion. The hydrological model exists in version with 100m and 500m, where we use results from the 500m version, as mentioned in line 233. In line 356, we refer to the resolution of the hydrologeological model – which is 100 m. The same hydrogeological model is used as basis for both the 500m and 100m version of the

hydrological model (as applies for much of the other input data; both the 100m and 500m versions of the hydrological model share the same input; the data then just is resampled internally by the modelling software to the actual hydrological model resolution. To clarify this, we added a sentence where the 500m and 100m versions of the hydrological model are mentioned, see lines 242-244.

L341ff: If you include these data, cite it here!

**Reply**: the references of the data has been added to the manuscript.

L470ff: Since I could not access your data, I could also not check your data description readme. First of all, you could and should refer to this readme here.

**Reply**: data description file has been mentioned in the first sentence, see line 476.

L473: Tables 2-5 (plural)

**Reply**: the typos has been corrected.

L476: I guess shapefiles are plural too.

**Reply**: the typos has been corrected.

L486f: I am still a bit unsure about the strong emphasis on groundwater – which is not a standard feature in other CAMELS publications. Moreover, you have not used a ML model to simulate the ungauged basins but a "physics-based grid model" which is also something "special". I suggest clarifying these specifics very early in the manuscript as part of the speciality of Denmark (as all CAMELS datasets have regional specialities) and to make it tangible as an interesting addition.

**Reply**: yes, we have added some sentences to explain why we want to have simulated groundwater dynamics in the dataset (see Lines 106-113).

We have developed ML models to simulated discharge in all catchments in another study (10.5194/hess-28-2871-2024), which has been cited in the current manuscript to explain the benefit of involving simulated variables, such as simulated streamflow, soil moisture, and shallow groundwater levels for streamflow modelling with Long Short-term Memory (LSTM) model (see lines 98 -101).

In the summary you give the quite generic statement, that your CAMELS dataset can "assist machine learning studies" – sure, this is exactly why CAMELS and more, so CARAVAN came into existence. But with your addition about

groundwater and the MIKE-SHE* references (since I could not access the data: are they existing for the catchments with observations as well?) this could be pointed out more specifically?

**Reply**: We have revised the conclusion/summary section of the manuscript. The reasons for including DK-model simulations are also discussed in this section; please refer to lines 486–501.

I am sorry to come with these points at a late state of the process. I hope it will be very easy for you to still do these small corrections before the Christmas break to have the manuscript swiftly accepted thereafter. If you disagree, maybe there is another minor detail which should be clarified so that readers like myself do not derail during reading?

**Reply**: We appreciate your comments on the manuscript and the dataset, they really helped us to improve the quality of the work.